# Research on Urban Public Green Space Planning Based on Taxi Data: A Case Study on Three Districts of Shenzhen, China

**Quanyi Zheng [1]**, **Xiaolong Zhao [1,*]** and **Mengxiao Jin [2]**

1   Key Laboratory of Cold Region Urban and Rural Human Settlement Environment Science and Technology, Ministry of Industry and Information Technology, School of Architecture, Harbin Institute of Technology, Harbin 150001, China; zhengquanyi@sina.com
2   School of Environment, Harbin Institute of Technology, Harbin 150001, China; 11649014@mail.sustc.edu.cn
*   Correspondence: zhaoxiaolong@hit.edu.cn

**Abstract:** Urban public green space (UPGS) plays an important role in sustainable development. In China, the planning, classification, and management of green spaces are based on the Standard for Classification of Urban Green Space (SCUGS). However, limitations to the UPGS exist due to the over-emphasis on quantitative standards and insufficient consideration of the actual access mode of residents. Though the taxi trajectory data are widely selected to study public service facilities, its adoption in UPGSs research remains limited. Based on the case of UPGSs in the three districts of Shenzhen, we used the taxi (including cruise taxis and Didi cars, which are like Uber) trajectory data to investigate the spatial layout and the allocation of management resource of the UPGSs from the spatial interaction perspective. By rasterizing and visualizing the percentage of pick-up and drop-off points in the UPGSs' buffer, the service scope of UPGSs was defined, which reflected the spatial distribution and activity intensity of the visitors. Then, an unsupervised classification method was introduced to reclassify the twenty two municipal parks in the three districts. Compared to the traditional planning method, the results show that the service scope of the same type of UPGS in the traditional classification is not the same as the one obtained by the study. Visitors to all UPGSs are distributed as a quadratic function and decay as the distance increases. In addition, the attenuation rates of the same type of UPGSs are similar. The findings of this study are expected to assist planners in improving the spatial layout of UPGSs and optimizing the allocation of UPGS management resources based on new classifications.

**Keywords:** urban public green space (UPGS); service scope; classification; urban planning; spatial interaction; taxi data

---

## 1. Introduction

As a natural resource in high-density cities, urban green spaces play an important role in improving human health and quality of life [1,2]. Urban green spaces can also mitigate urban heat islands, which have a negative impacts on the urban population [3,4]. Urban green spaces provide the residents with an ideal place for leisure activities and social interactions, as well as enhancing human health by reducing urban noise and controlling air pollution [5]. Urban green spaces can be divided into private and public green spaces based on different ownership. Private green spaces refer to outdoor facilities in private residences or non-residential green spaces that require payment, such as golf courses. The accessibility of private green spaces is not restricted through permission of the owners. In contrast, urban public green spaces (UPGSs) represent publicly owned green spaces that can be accessed freely without payment [6]. UPGSs include natural places (such as forests and parks)

and artificial green spaces (such as riverside green belts and squares) [6,7]. To assist city planners and managers in evaluating and analyzing the characteristics of UPGSs, scholars have completed many studies on this area, with a focus on areas such as accessibility [8,9], service scope [10], and equity issues [10]. An accurate assessment of the UPGSs is crucial for reasonable urban planning [11,12]. Therefore, UPGSs are receiving increasing attention in numerous research fields such as ecology and geography [13–15].

The core view of these studies is to evaluate, from spatial perspective, the connection between supply and demand locations. The connection between the demand locations and supply locations is essentially a spatial interaction that can be quantified by human mobility, goods, and capital [16]. As we all know, the relationship among the urban functional structure, spatial pattern, and visitors is inseparable. However, in China, the planning, classification, and management of green spaces are in accordance with the Standard for Classification of Urban Green Space (SCUGS) set up by the Ministry of Construction in 2018. SCUGS is based on the service radius and rigid indices, such as the green space area per capita and the green space ratio, to guide urban planning. SCUGS also has some limitations which have caused problems in green space planning. A typical problem is that quantity of green spaces was often emphasized, but the rational layout was not fully considered [17]. In some circumstances, planners and policymakers study UPGSs using a top-down approach and, therefore, do not fully consider the actual access mode of residents. The traditionally planned service scope is a standard circle, while the actual UPGSs' service scopes are irregular shapes [18]. At the same time, it does not consider the actual resistance of the road and the influence of distance attenuation. This results in a large difference between the actual service scope and the plan [19]. The classification of the green spaces and the allocation of management resources are also based on SCUGS. For example, in the traditional classification, the classification criteria for amusement parks and other specialized parks only emphasizes that the quantitative indices of the green space ratio should be greater than or equal to 65%. Their service scopes and the allocation of management resources are then divided into the same category. However, the actual service scopes and management resources of UPGSs are different, and they are affected by many factors [18]. This method results in an imbalance in the allocation of green space resources, which leads to some UPGSs with service overload and some UPGSs that are not fully utilized or idle. Meanwhile, due to the insufficient consideration of the actual access mode of visitors, this method cannot reflect the actual space interaction and service scopes of UPGSs.

In addition, most of the traditional methods rely on the static data, such as census data, travel surveys, and access records. However, it is difficult to obtain traditional data for surveys and research as doing so requires a lot of labor, material, and financial resources. With the in-depth application of big data in urban planning, a massive volume of data on human movement and flow can be collected to quantify spatial interactions between places [20]. Scholars used spatial interaction information extracted from big data to study urban space. Based on 500,000 pieces of social media data, Liu et al. analyzed the basic patterns of travel and spatial interactions from the human mobility patterns perspective and verified the law of distance attenuation effects [21]. Roth et al. used personal subway check-in data in London to reveal the structure and organization of the city while providing a method for modelling the flow in urban systems [22]. Ratti et al. proposed a new fine-grained method of region partitioning based on data recorded from telephones [23]. Ronghao et al. quantified the distribution and activity intensity of people in cities based on Baidu POI (point of interest) data and proposed evaluation indicators of service stress to improve the spatial layout of green spaces [24]. Liu et al. used the unsupervised method to incorporate spatial interaction models into land-use classifications based on Shanghai taxi data [25]. Kong et al. studied the spatial interactions and service scopes of Beijing hospitals based on Beijing taxi data and verified the effect of the law of distance attenuation on the hospitals [16]. Although many scholars use taxi trajectory data to study public service facilities, few scholars use taxi trajectory data to describe UPGSs. Visiting UPGSs in taxi travel mode is a crucial research because, at least in some cities in China, the proportion of people visiting UPGSs in taxi travel



mode is not small [26]. Therefore, this work studied the relationship between people and cities based on big data and a "human-oriented" perspective.

The study took the three districts of Shenzhen as an example and used taxi trajectory data to investigate the spatial layout of UPGSs and the management resource allocation from the spatial interaction perspective. The method was used to compensate for the limitations of traditional planning methods in China. In order to achieve this goal, the following problems needed to be solved: (1) How can we use trajectory taxi data to quantitatively analyse the distribution of people and the intensity of human activities to describe the UPGSs' service scopes and the demand degree for UPGSs? (2) Are the traditional classification of the service scope and spatial interaction of the same type of UPGS the same? (3) How can we reclassify the UPGSs based on the source of visitors from the spatial interaction perspective?

## 2. Research Data Description and Processing

### 2.1. Study Area and Municipal Parks

Shenzhen is located in the Southern part of the Guangdong province in China, which borders Hong Kong and is an emerging immigrant city. The issues of UPGS planning in high-density cities, such as Shenzhen, are common in China and most other countries. Therefore, the results apply not only to other cities in China but also to similar cities in Asia, Europe, and the United States. The Nanshan, Futian, and Luohu districts are the central districts in Shenzhen as the administration, culture, finance, and information centers of the city (Figure 1). They are adjacent to the Longhua and Longgang districts to the north, the Baoan district to the west, the Yantian district to the east, and Hong Kong to the south, covering a total area of 339.55 km$^2$. The population of the three districts is around 1.36 million, 1.50 million, and 1.40 million, with capita public green area about 16.98 m$^2$, 22.52 m$^2$, and 17.0 m$^2$, which are more than the 10–15 m$^2$ recommended by the World Health Organization (WHO).

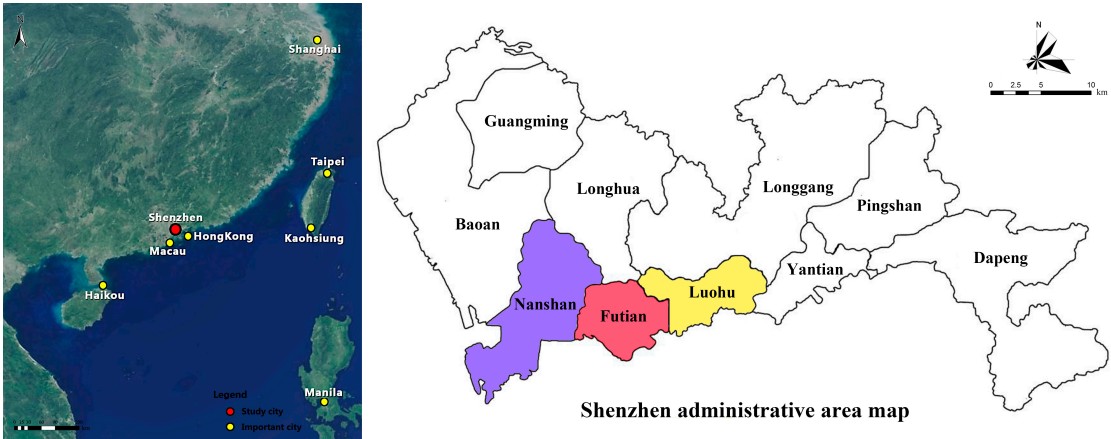

**Figure 1.** Location of study area: Nanshan District, Futian District and Luohu District in the city of Shenzhen in Guangdong Province, China.

To measure the spatial interaction strengths, the study area was discretized into 1460 500 × 500 m square cells. Previously, 1000 × 1000 m and 200 × 200 m cells were tested separately. Given the scale, in the former test, one cell contained two or more green spaces; in the latter test, one green space was distributed in several cells. All trips extracted from the taxi trajectory data were aggregated based on these 500 × 500 m cells. Municipal parks are typical UPGSs, which are funded by the government and are free for urban residents. As shown in Figure 2, the twenty two municipal parks (Table 1) were selected in three districts of Shenzhen.

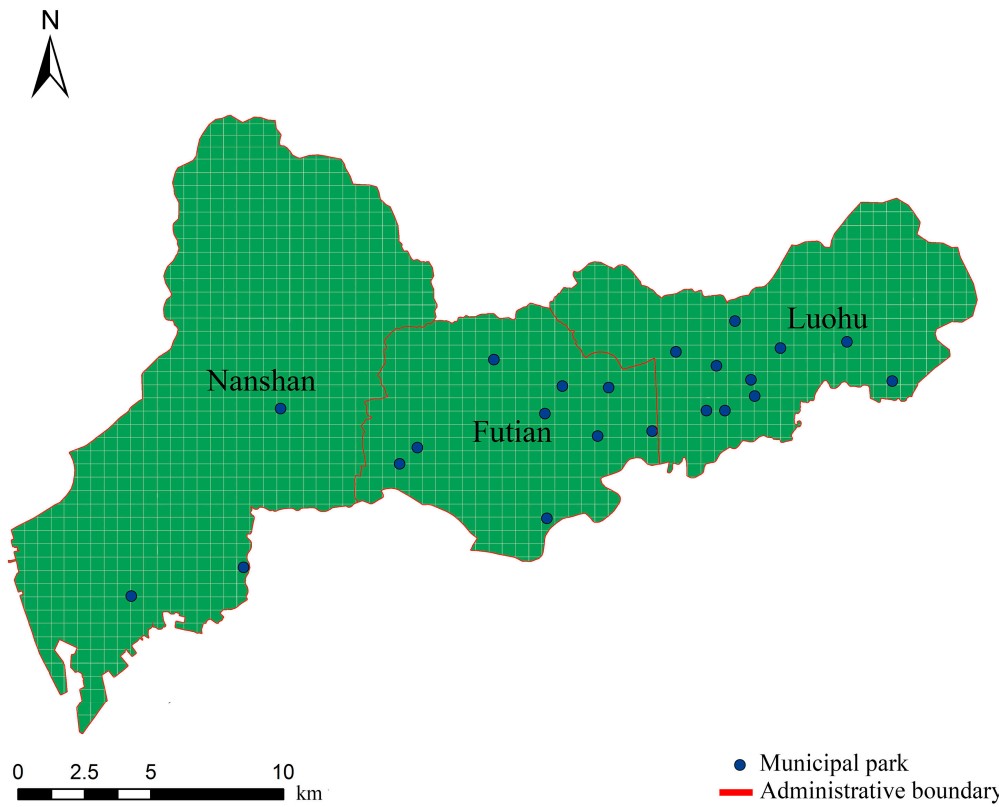

**Figure 2.** Spatial distribution of municipal parks in the study area.

**Table 1.** Twenty two UPGSs in three districts.

| Name of UPGS | Longitude | Latitude | District |
|---|---|---|---|
| Dananshan park | 22.49724852 | 113.8921728 | Nanshan district |
| Dashahe park | 22.5600313 | 113.9567846 | Nanshan district |
| Shenzhen bay park | 22.50739808 | 113.948838 | Nanshan district |
| Children's playground | 22.54548378 | 114.0064251 | Futian district |
| Caitian park | 22.56559133 | 114.0596258 | Futian district |
| Bijiashan park | 22.56418573 | 114.0764316 | Futian district |
| Flower expo park | 22.54123803 | 113.9984341 | Futian district |
| Huanggang park | 22.52111487 | 114.0525387 | Futian district |
| Meilin park | 22.5731751 | 114.0345282 | Futian district |
| Lizhi park | 22.54919824 | 114.097807 | Futian district |
| Lianhuashan park | 22.55719315 | 114.0557774 | Futian district |
| Shenzhen Central Park | 22.53531121 | 114.0701766 | Futian district |
| Donghu park | 22.56373323 | 114.1428801 | Luohu district |
| Honggang park | 22.57809532 | 114.1093117 | Luohu district |
| Honghu park | 22.56577483 | 114.1154877 | Luohu district |
| Weiling park | 22.58626246 | 114.1235435 | Luohu district |
| Xiantong Sports Park | 22.56486056 | 114.1808784 | Luohu district |
| People's Park | 22.55628535 | 114.111858 | Luohu district |
| Cuizhu park | 22.56598516 | 114.1269159 | Luohu district |
| Children's park | 22.55621535 | 114.119288 | Luohu district |
| Luofang park | 22.55485458 | 114.1407913 | Luohu district |
| CAS Xianhu botanical garden | 22.57858608 | 114.1655407 | Luohu district |

*2.2. Data*

2.2.1. Urban Public Green Space Data

According to the data released by the Shenzhen Municipal Administration, the content information was obtained for the twenty two municipal parks in the Nanshan, Futian, and Luohu districts. The latitude and longitude information of the UPGSs required for the study was contained in UPGSs' POI (point of interest) data collected from the Gaode map (Chinese web mapping provider). POI data is a new type of data for location services. Since the POI data is an abstract point without area or volume, it is necessary to link the area of the UPGSs with the electronic map data. The geographic information of the twenty two municipal parks in the three districts was obtained from the Gaode map. As the UPGS data required spatial verification, the study combined the spatial location with the POI data and verified it by manual visual interpretation. The result was an exact match.

2.2.2. Taxi Trajectory Data

The processed and analyzed taxi trajectory data can reflect the intensity and spatial distribution of urban residents. Taxi trajectory data is a typical, passive, and group perception data. It is generated when visitors use the various information infrastructures of the city in an unconscious state. The study extracted trajectory data from over 16,000 taxis (including cruise taxis and Didi cars, which are like Uber) in Shenzhen for three consecutive months (July–September 2017). As according to the "Analysis of Monthly Congestion Trends of Major Cities in 2017" [27], the peak period of congestion in major cities in China begins to rise in July and peaks in September. During this period, the urban transportation and tourism activities are the most representative. In addition, as the summer vacation is between July and September, the travel frequency of students is higher than usual. The GPS of the vehicles collected the real time position every 20 seconds and a total of 56,597,588 position records were acquired. The dataset contains the taxi's unique number, longitude, latitude, speed, direction, positioning time of occurrence, and status (Table 2). Before the data was analyzed, the taxi trajectory data needed to be pre-processed to exclude off-road or erroneous data and to delete duplicate or defective GPS records. Through this pre-processed data, the locations of the visitors' origin (or destination) were extracted as well. Each route from the origin $(x_o, y_o, t_o)$ to the destination $(x_d, y_d, t_d)$ was converted into vector data, where "x" and "y" represent the position of the pick-up point (PUP) and the drop-off point (DOP) and "t" represents the time when the pick-up and drop-off behavior occurred. The spatial interaction between the locations can be represented from the extracted data.

**Table 2.** Taxi data sample.

| Field Name | Types | Remarks |
| --- | --- | --- |
| Id | LONG | Serial number |
| Taxi_id | TEXT | Taxi unique number |
| Longitude | FLOAT | longitude |
| Latitude | FLOAT | latitude |
| Speed | DOUBLE | Instantaneous speed (km/h) |
| Angle | DOUBLE | Azimuth (degrees) |
| Unix | LONG | Unix timestamp |
| Status | LONG | Passenger status (1 guest, 0 guest) |

2.2.3. Other Data

Other data includes road network data and administrative boundary data. The road network data was obtained from the website of Open Street Map (OSM) in 2017. The administrative areas of the three districts were derived from the GIS data of the 2017 Shenzhen land use vector data map.

## 2.3. Taxi Travel Mode Related Investigation

Taxi travel mode is one of the important modes of visiting UPGSs in some cities in China. Tianqing et al. [26] selected eleven well-known UPGSs in Shanghai, and the study was completed in the sunny days of March–June 2009. A total of 15 surveys were conducted, and 588 valid questionnaires were obtained. The study found that four UPGSs had a higher proportion of non-sustainable travel mode (taxis) than the other parks, and the proportion are 24%, 37%, 40%, and 45%, respectively. The result shows that the proportion of taxi travel mode is not low. In addition, in order to demonstrate the proportion of taxi mode travel, we selected six municipal parks in the three districts of Shenzhen for a questionnaire survey. When studying UPGSs with similar distances, one should pay attention to the difference in size and time of completion to increase the representativeness of the sample. The investigation of the municipal parks was concentrated in the sunny days of October 2018. We collected data on the weekend when the visit rate was high. The questionnaire included the travel mode and source of the visit. A total of five surveys were conducted, and 1021 valid questionnaires were obtained. The survey found that 11.36% of visitors chose to visit the UPGSs in taxi travel mode. The growth of the taxi travel mode stems from the development of the network cars. However, the most popular modes of travel were still subway (32.81%) and bus (19.59%), followed by walking (16.16%) and bicycles (11.66%). Private cars were only 8.42%. Though the taxi travel mode has grown faster, the proportion of taxis is still small compared to other travel modes. Due to the relatively small proportion of taxi users, the study has some limitations. However, considering the population has a large base in Shenzhen, the taxi travel mode is still very valuable for study.

## 2.4. Calculating the Pick-Up and Drop-Off Points

Based on the average width of urban roads in China, a multi-layer central ring buffer at a specified distance of 30 m around the UPGS was constructed. It was used to distinguish the DOPs and PUPs around the UPGSs (Figure 3). The graph shows that the percentage of PUPs and DOPs in each ring is approximately linear at the beginning, appears to be stable at 180–210 m, and finally increases from 240–270 m to 270–300 m. In addition, because the actual boundaries of the UPGSs were irregular shapes—and the data were difficult to obtain—if the actual boundary of the UPGS is considered, other factors such as its internal road network should be taken into account as well. The complexity and scale of the study thus increased. The shape core of the UPGSs was chosen because many scholars [28–30] used it to study spatial accessibility and equity. This is a mature research method, and the error of the result calculated by this method is within an acceptable range. As shown in Figure 3, the percentage tends to be smooth starting at 200 m, which is a stable interval. A threshold of 200 m was used, which is an acceptable walking distance for a person. A study found that the distance between the PUPs (or DOPs) of the taxi and the actual place of departure (or the place of arrival) is mostly around 200 m [16]. Since the environment around municipal parks varied, it was actually impossible to completely distinguish between different situations. In order to include more possibilities, we chose a larger buffer. Following this, we drew a 200 m buffer for each UPGS, which ensured that there was no interaction between each buffer (Figure 4). Though the 200 m buffer caused some noise, there were many ways to eliminate them. For example, because the taxi data were all-day data, we used the time dimension to distinguish the data from the morning and evening peaks, as well as the data for meal times to eliminate noise from residential areas, schools, and restaurants. In addition, commercial, medical, and other functions near the UPGSs are interactive, and some proportion of visitors takes the opportunity to go to the UPGSs for leisure. In summary, although the data had some noise, the sample size was large, and the proportion of noise in the sample size was very small. Therefore, the error was within an acceptable range. Within each buffer area, the DOPs or PUPs of trips that fell within the area were detected. The travel trajectory information of these urban residents was assumed to be the data of visitors visiting the UPGSs. In short, based on the vector mentioned above, from $(x_o, y_o, t_o)$ to $(x_d, y_d, t_d)$ for each visitor in the taxi dataset, an OD (origin-destination) linkage was built. Subsequently, each cell was given a weight based on the number of taxis that fell

within it. The weight of a cell was represented by the number of trips between the UPGS and a location inside the cell. Finally, a 22 × 1460 matrix was created. Within this matrix, the rows represented the UPGSs, the columns represented each cell, and the intersections of the rows and columns represented the number of taxi trips between the UPGSs and the cells.

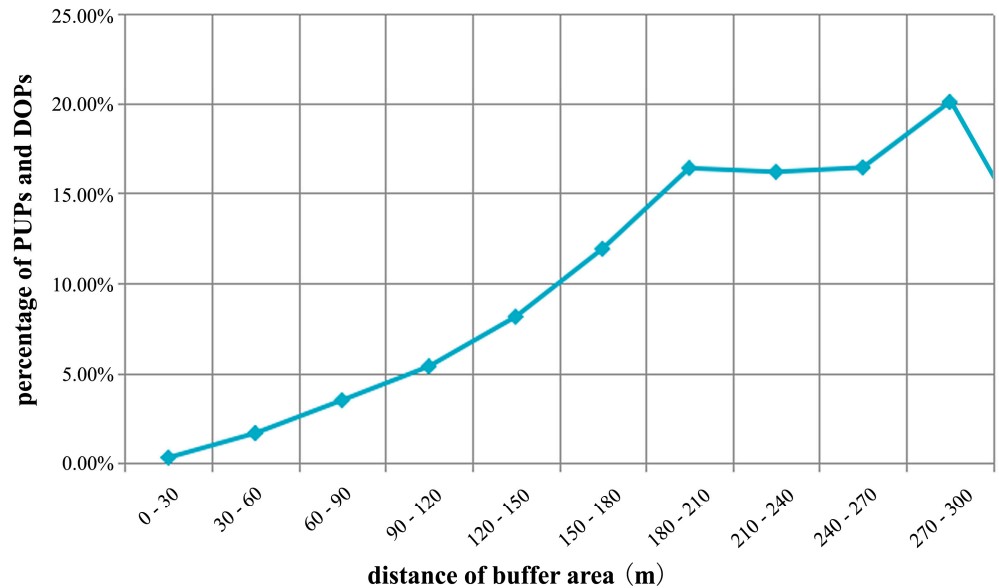

**Figure 3.** Percentage of pick-up points (PUPs) and drop-off points (DOPs).

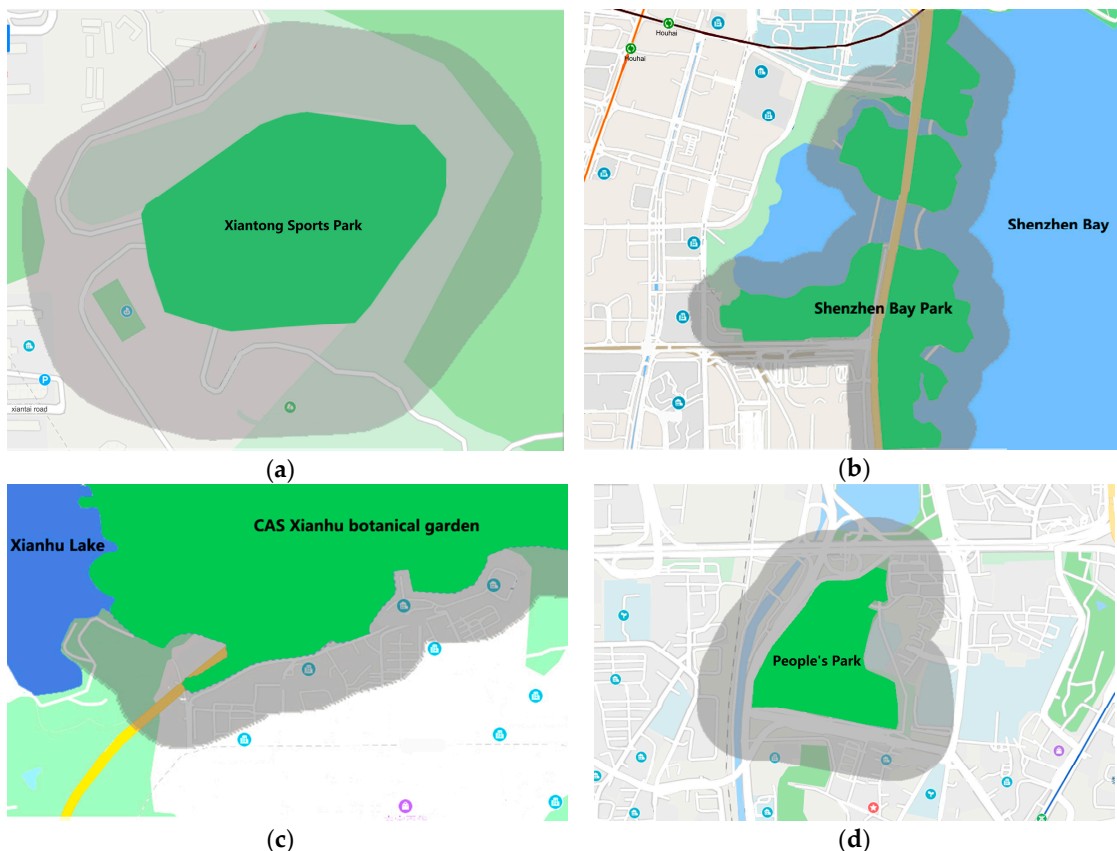

**Figure 4.** Representative municipal parks. (**a**) Xiantong Sports Park; (**b**) Shenzhen Bay Park; (**c**) CAS Xianhu Botanical Garden; (**d**) People's Park.

## 3. Methods

The study methods include the analysis of the UPGSs' service scopes and the classification of the UPGSs from the spatial interaction perspective.

### 3.1. Analysis of the UPGSs' Service Scopes

The service scope is an important spatial characteristic. It reflects the attractiveness of the UPGSs, and it is related to the spatial distribution of residents. The criteria for residents to choose a UPGS depends on the spatial distance between them [31]. The ideal way to study the behaviour of space selection is to measure the flow of residents visiting the UPGSs. The data can be obtained from government statistics, but it is not public. Therefore, scholars use the Huffer model or the 2SFCA (two-step floating catchment area) model to assess the spatial distribution of residents and the service scopes of the UPGSs [6,32]. Even if the survey data is obtained, they have some limitations such as high cost in terms of time, manpower, finance, insufficient sample size, and low measurement efficiency. At the same time, the quality of the survey data, such as low spatial accuracy and large time granularity, cannot be ignored. On the contrary, the advantages of big data are obvious and include low-cost acquisition, sufficient sample size, and efficient measurement. Comparing with traditional data, big data are more accurate and objective in characterizing urban public facility (such as the UPGSs) characteristics [33]. The study is based on a large amount of human mobile data, which is extracted from taxi data, then combined with POI data, electronic map data, OSM road network data, and other data to calculate the service scope. In order to display the UPGSs' service scopes, the spatial distribution of residents of the twenty two UPGSs were visualized and rasterized. The service scope value represents the activity intensity of residents visiting the UPGSs within the 500 $\times$ 500 m cell. It describes the spatial distribution of residents and also reflects the residents' demand for UPGSs.

The spatial distribution of residents visiting the UPGSs was examined based on the spatial interaction using the visual interpretation method. Distance is an important factor in spatial interaction, and, thus, spatial interaction can be quantified by distance attenuation [16]. The spatial interaction will decrease with increasing distance. This effect is called the distance attenuation effect. The distance decrease function can be used to characterize the effect of the distance. Commonly used distance attenuation functions include exponential functions, power functions, Gaussian functions, and the quadratic function. The attenuation of the distance will affect residents visiting the twenty two UPGSs. Therefore, the decay rate was chosen as the comparison factor, since it can be used to quantify the role of distance in the spatial interaction between different UPGSs and visitors. Furthermore, it is an important parameter in a public facility accessibility calculation model [34]. The decay rate index directly controls the speed of the attenuation.

### 3.2. Classification of the UPGSs from the Spatial Interaction Perspective

Different public facilities have different service scopes, and different visitors have different travel distances [16]. Therefore, the UPGSs were reclassified based on the source of the visitors. The traditional UPGS classification is based on SCUGS. It is divided into comprehensive parks, community parks, theme parks, linear parks, and street green spaces. According to the above, community parks, linear parks, and street greens are small size parks and not municipal parks. Therefore, they are not within the study scope. The comprehensive parks are only divided into urban parks and regional parks. However, the theme parks are divided into seven types of parks, such as children's parks, zoos, botanical gardens, historical gardens, landscape parks, amusement parks, and other specialized parks. In order to find the limitations of the traditional UPGS classification, the study selected six parks from the twenty two UPGSs and divided them into three groups. In each group, two UPGSs of the same type in the traditional classification were selected. Then, the service scope and distance attenuation rate of the two parks in the three groups were compared and analyzed.

The first two parks were urban comprehensive parks, the second two parks were children's parks, and the third two parks were botanical gardens.

The most critical technique for analyzing trajectory data is cluster analysis [35], which is the most commonly used research method in data mining. It is an unsupervised learning method and does not rely on pre-defined classes. By grouping similar things together, objects in the same set can show similar properties. The K-means algorithm is very effective in dealing with massive data, especially for numerical data processing. It is very sensitive to anomalous data and cannot effectively partition non-convex data [36]. Therefore, the study used the K-means clustering method [37], as it was a classical algorithm which had the advantages of fast and easy calculation [16]. The algorithm maintains scalability and efficiency when working with large data sets. The UPGSs were clustered based on the distribution of the residents from the result in Section 3.1. Therefore, the UPGSs were divided into K classes by the K-means clustering method. However, the new classification K class was different to the traditional classification method. Its classification was based on spatial interaction information (distance attenuation) extracted from taxi data. The new classification considered the access mode of visitors and reflected the UPGSs' objective supply and demand situation. The specific process is as follows:

1.  The study assumes that the sample data set (all observations) is divided into K classes and arbitrarily selects *k* objects from *n* data objects as the initial cluster center;
2.  The distance between each object and these central objects is calculated according to the mean value (center object) of each cluster object and the corresponding objects according to the minimum distance are re-divided;
3.  The mean (central object) of each (with change) cluster is recalculated;
4.  The standard measure function is calculated and when certain conditions, such as function convergence, are met, the algorithm terminates. If the conditions are not met, steps 2 and 3 are repeated.

The calculations are based on the following formula:

$$E = \sum_{i=1}^{k} \sum_{x \in C_i} \|x - \mu_i\|^2 \tag{1}$$

where *x* represents observations and $\mu_i$ is the average of the observations in cluster $C_i$. To obtain better classification results, it was necessary to repeatedly compare the classification results under different *k* values. By comparing the classification results under different *k* values, when the number of UPGSs types was 3, the criterion function reached the minimum, and the classification result was more reasonable.

## 4. Results and Analyses

### 4.1. UPGSs' Total Service Scope and the Representative UPGSs' Service Scope

As shown in Figure 5, the visualized taxi data can accurately reflect the UPGSs' service scopes and can also describe the spatial distribution of residents and the actual activity intensity of residents. First, the service scope value in the Northeastern part of Nanshan District, the Northern part of Futian District and the Eastern part of Luohu District were close to 0, because these areas were mountain and natural landscape reserves. In addition, the service scope value in the Southwestern part of Nanshan District was very low because the area was a dock. The intensity of resident activity in these areas was low, which proved that their demand for UPGSs was low. The visualized taxi data not only reflects the spatial interaction between UPGSs and residents but also indirectly reflects the actual geographic information in the study area. Secondly, the service scope value in the central Nanshan District, the central Futian District, and the Western part of the Luohu District was very high, because these areas had a large number of residential areas with a high population density. In particular,

the central part of Nanshan District and Futian District had the highest values, as they were the center of science, technology, finance, and information in Shenzhen, and they also had the highest quality UPGSs. Therefore, the activity of people in these areas was intensive and their demand for UPGSs was high, so the UPGSs in these two areas were very attractive. Finally, the road network in the high-value area was better than the other areas, and the accessibility of these areas was better.

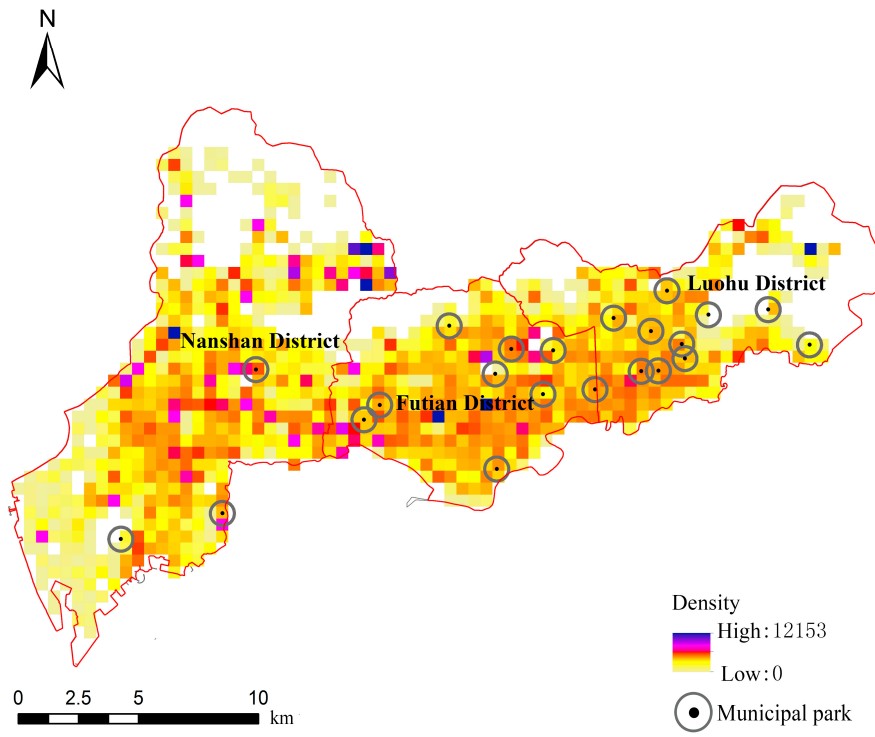

**Figure 5.** The twenty two urban public green spaces' (UPGSs') service scopes.

To study the service scopes of UPGSs in detail, six UPGSs were selected from the twenty two UPGSs as examples: (a) Bijiashan Park; (b) Shenzhen Bay Park; (c) Children's Playground; (d) Children's Park; (e) Flower Expo Park; and (f) CAS Xianhu Botanical Garden (Figure 6). Six UPGSs can be divided into three groups. First, Bijiashan Park and Shenzhen Bay Park are comprehensive municipal parks and famous tourist attractions in Shenzhen. Second, Children's Park and Children's Playground are theme parks. Third, Flower Expo Park and CAS Xianhu Botanical Garden are research-oriented UPGSs, but they are also open to the public.

The service scopes of the six UPGSs were significantly different. UPGSs 6(a) and 6(b) in the first group were comprehensive parks, and, because of their different locations, their service scopes were different. Due to the geographical location, the service area boundary of UPGS 6(b) presented an irregular shape, which was consistent with the results for the above overall UPGS service scope. The service scope of UPGS 6(b) was greater than that of 6(a), because UPGS 6(b) was rich in services (such as sports, cultural performances, and management), while UPGS 6(a) was relatively simple. In addition, the traffic road network around UPGS 6(a) was not perfect, and the service scope value was slightly lower than 6(b). UPGSs 6(c) and 6(d) in the second group were both theme parks. The service scope of UPGS 6(c) was much larger than that of 6(d). The reason for this was that the service facilities and reputation of UPGS 6(c) were better than 6(d). Furthermore, its location was at the junction of the two districts and will attract residents of both districts. As can be seen from the boundary of the UPGS 6(c) service scope, it was clearly affected by the Northern mountains. UPGSs 6(e) and 6(f) in the third group were both botanical gardens. Their service scopes were quite different. The reason was that UPGS 6(e) was an exhibition-oriented green space, and it was also an important tourist attraction in Shenzhen. It was very attractive and has many visitors. UPGS 6(f) was a research-oriented green space.

Its location was also relatively remote, and visitors mainly came from nearby. Although the service scope of UPGS 6(f) was small, the service scope value was high. It showed that the accessibility of the area was good and there was strong interest from the visitors. Through the above comparative analysis, it was found that the actual service scopes of the same type of UPGSs were not the same, and the activity intensity of the people around them were also different. Geographical factors had a significant impact on the service scope. Moreover, most visitors were concentrated in the area surrounding the UPGSs, which indicated that people often chose to visit nearby UPGSs. However, some residents may choose a farther UPGS for special needs (such as leisure activities with children).

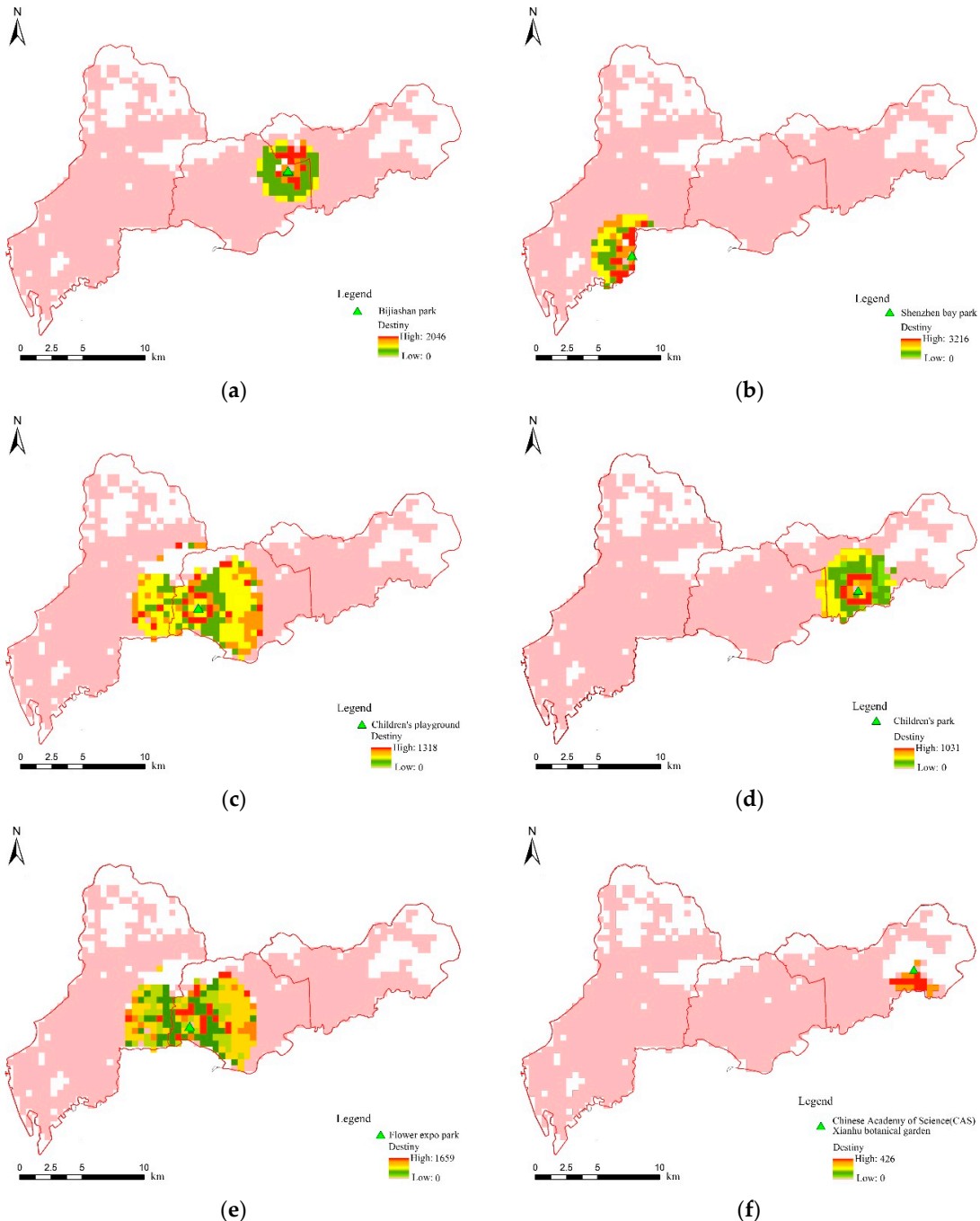

**Figure 6.** UPGS service scope. (**a**) Bijiashan park; (**b**) Shenzhen bay park; (**c**) Children's playground; (**d**) Children's park; (**e**) Flower expo park; and (**f**) CAS Xianhu botanical garden.

The effect of the trip distance on visits to the UPGSs was examined by discussing the relationship between the interaction strength and distance (Figure 7). The x–axis represents distance (km), and the y–axis represents the number of PUPs (or DOPs) under a certain distance. As shown in Figure 7, the number of PUPs (or DOPs) increased and reached a peak when the distance was about 800 m. After that, the number of visitors decreased with distance. The distribution of the UPGS visitors with distance first increased and then decreased. From the curve fitted in Figure 7, the highest degree of fitting was the quadratic function ($R^2$ = 0.86), which was consistent with the general law of people's service utilization [38].

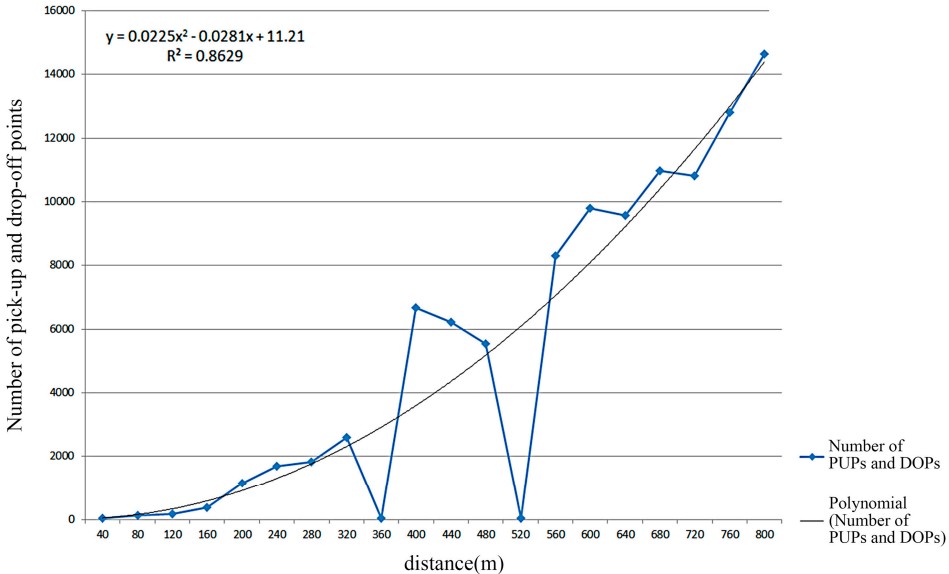

**Figure 7.** The distance decay function of the twenty two UPGSs visits.

Though we described the overall distance attenuation of the twenty two UPGSs, different UPGSs may present different distance attenuation effects due to different factors (such as location and service content). Therefore, the distribution of visitors of the five UPGSs were plotted to represent the effect of distance (Figure 8). As the CAS Xianhu Botanical Garden had too few PUPs (or DOPs) within 800 m, it was impossible to count. As shown in Figure 8, the distance factor affected the distribution of visitors to the five UPGSs. However, the decay rates ∣a∣ (absolute value of a) of the distance decay functions of the quadratic function represented different attenuation velocities. The larger the ∣a∣, the smaller the opening of the curve of the quadratic function and the faster the attenuation. The distance attenuation trends reflected by UPGSs 8(a)–(e) were the same. Except for UPGS 8(b), the other UPGSs were increasing within the 800 m scope. UPGS 8(b) dramatically increased and decreased within the 800 m scope. It was an individual and accidental case, but it also reflected the whole process of the distance attenuation effect. For the same type of UPGS, the distance decay rates were similar. UPGSs 8(a) and 8(b) had similar distance attenuation rate distribution patterns of 0.0011 and 0.0044. Compared with UPGS 8(a), the distance attenuation effect of 8(b) was more obvious. The reason for this was that UPGS 8(b) faced the sea and the route to the park was shorter than to 8(a). UPGSs 8(c) and 8(d) were theme parks that had similar distance attenuation rates of 0.0005 and 0.0004. Their average was lower than the comprehensive park, which meant that visitors to these parks were not sensitive to travel distances. Visitors to this type of UPGSs were children and their family members. Therefore, distance was not the main factor, and content was the first factor. UPGS 8(e) also satisfied the fitting of the quadratic function. If the visitors had a demand for the service content, they would not choose the UPGSs nearby. The second factor to consider was distance, which took into account the time and cost. The spatial interaction between the visitors and the UPGSs were highly consistent with the actual situation.

### 4.2. New Classification of UPGS Based on Spatial Interaction

The classification results of the twenty two parks were shown in Figure 9. Most of the theme type UPGSs were divided into class 1, the mixed type UPGSs were divided into class 2, and most of the comprehensive type UPGSs were divided into class 3. Table 3 shows the results of the specific classification. The new classification reclassified the traditional classification of UPGSs based on the actual source of the population and broke the original classification criteria. The traditional classifications are numerous and the content is cumbersome. In particular, allocating management resources to the same type of UPGSs without considering the actual situation will result in unbalanced management resources. After the cluster analysis, the probability distribution of each type of UPGS was visualized (Figure 10). As shown, there was a tendency for distance attenuation in the distribution of visitors of each type of UPGS, but the attenuation rate was significantly different. Furthermore, to verify the relationship between the distribution of visitors and the distance for each type of UPGS, these distributions were fitted by a quadratic function, and the attenuation index was calculated; the distribution of visitors and distance decay rate ｜a｜ were combined to analyze various types of UPGSs. The first class was the theme UPGS, and its attenuation rate was −0.0061. Due to the diversity of services, this type of UPGS attracts many people every day. The second class was the mixed type of UPGS. The distribution presented significant distance attenuation, and its attenuation rate index was 0.0108. The third class was comprehensive UPGS. This type of the UPGS was single-function tourist attraction, and its attenuation rate was 0.0175. The distribution of visitors of such UPGS was significantly affected by the distance.

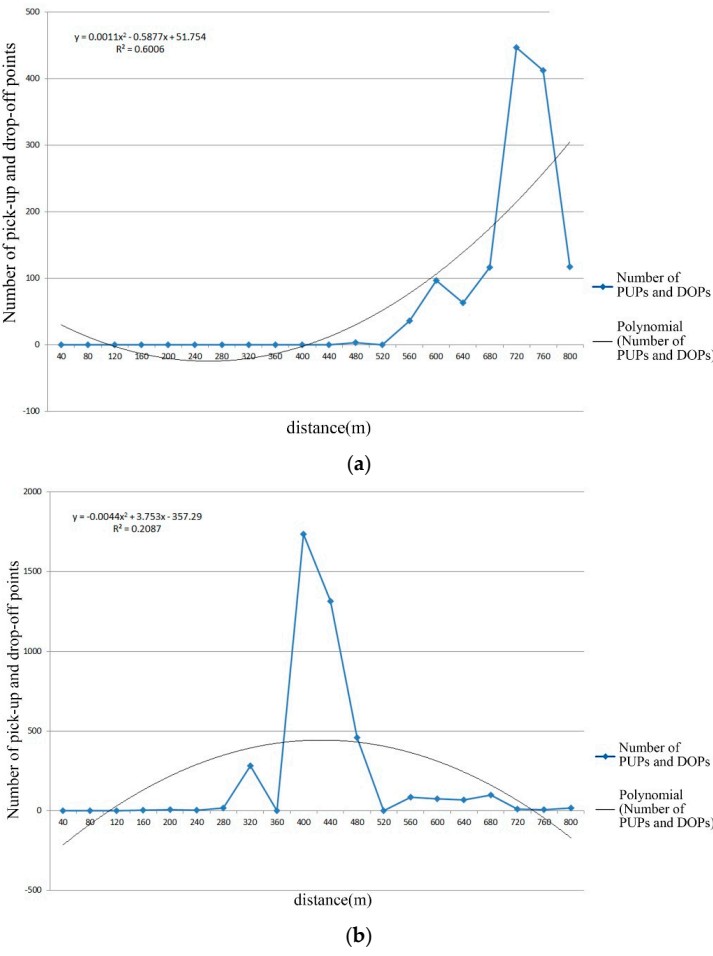

**Figure 8.** *Cont.*

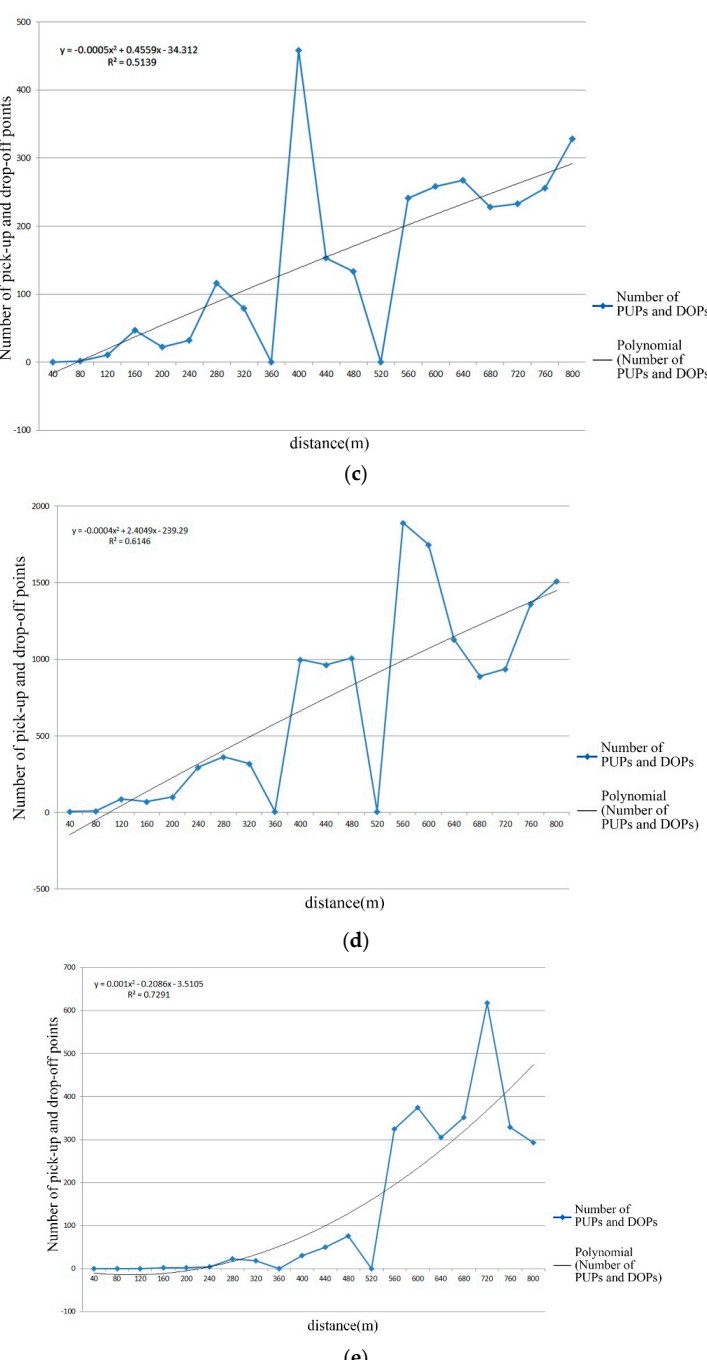

**Figure 8.** Distance distributions of typical UPGSs' visits. (**a**) Bijiashan park; (**b**) Shenzhen bay park; (**c**) Children's playground; (**d**) Children's park; (**e**) Flower expo park.

**Table 3.** Classification results of UPGSs.

| Cluster | UPGS Description | The Name of UPGSs |
|---|---|---|
| 1 | Theme UPGS as the main type | Children's park, Luofang park, CAS Xianhu botanical garden, Children's playground, Caitian park, Dananshan park, Dashahe park, and Shenzhen bay park. |
| 2 | Mixed type | Donghu park, Honggang park, Honghu park, Cuizhu park, Weiling park, Xiantong Sports Park, Bijiashan park, Meilin park, Flower expo park, and Huanggang park. |
| 3 | Comprehensive UPGS as the main type | People's Park, Lizhi park, Lianhuashan park, and Shenzhen Central Park. |

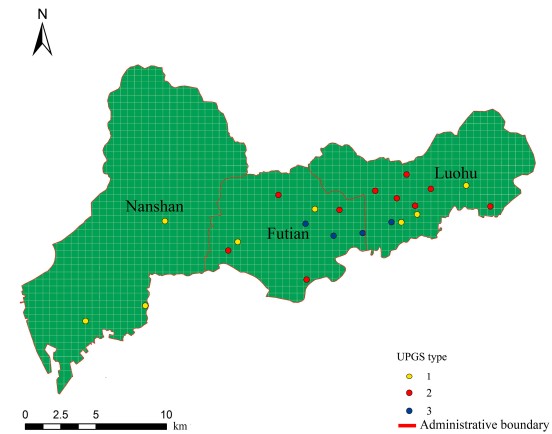

**Figure 9.** Classification result of twenty two UPGSs.

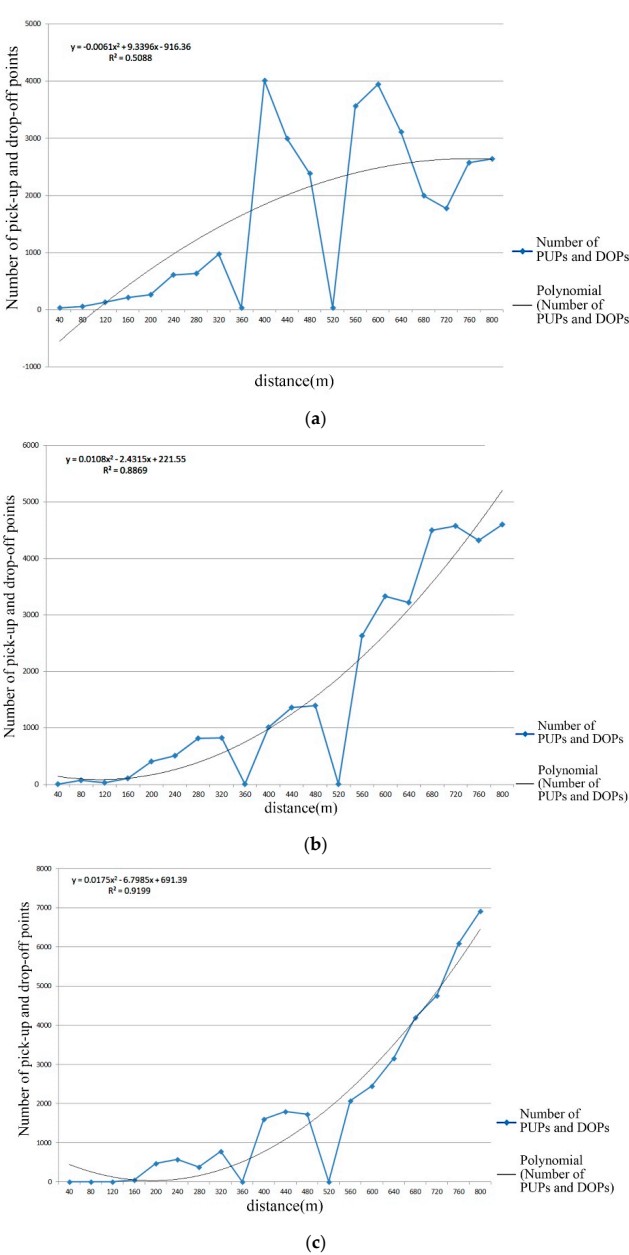

**Figure 10.** Distance distribution associated with each class of UPGS. (**a–c**) represent the clusters of UPGSs separately.

## 5. Discussion

This study was based on the spatial interaction information extracted from the taxi trajectory data. By rasterizing and visualizing the calculated data above, the service scopes of the UPGSs were quantified. The service scope not only describes the distribution of visitors and the intensity of human activity but also reflects the accessibility and actual geographic information of the UPGS. The above results verified that the service scope of the same type of UPGS in the traditional classification was not the same as the one obtained by the study. Then, is the traditional service scope the same as the service scope for big data visualization? Comparing with the traditional service scope, the new service scope objectively and accurately reflects the diverse characteristics of UPGSs and the relationship among the residents, UPGSs, and cities. In China, the traditional service scope is based on the SCUGP and is determined by the radius of service. However, the traditional service scope is a regular circle, and it ignores the actual road resistance and the actual distribution of the visitors. Take the comprehensive park as an example. As shown in Figure 11, compared to the traditional service scope, the actual service scope boundary was an irregular shape. The former did not take into account the effect of distance attenuation, and thus, it was evenly distributed. On the contrary, this study considered the effect of distance attenuation, which can describe its scope and the boundaries of the actual service scope. According to the above, the spatial interaction was quantified by the distance attenuation, and the attenuation rate index $|a|$ was used as a contrast factor. Therefore, the same type of UPGSs had similar attenuation rates, and the spatial interactions of the same type of UPGSs were similar.

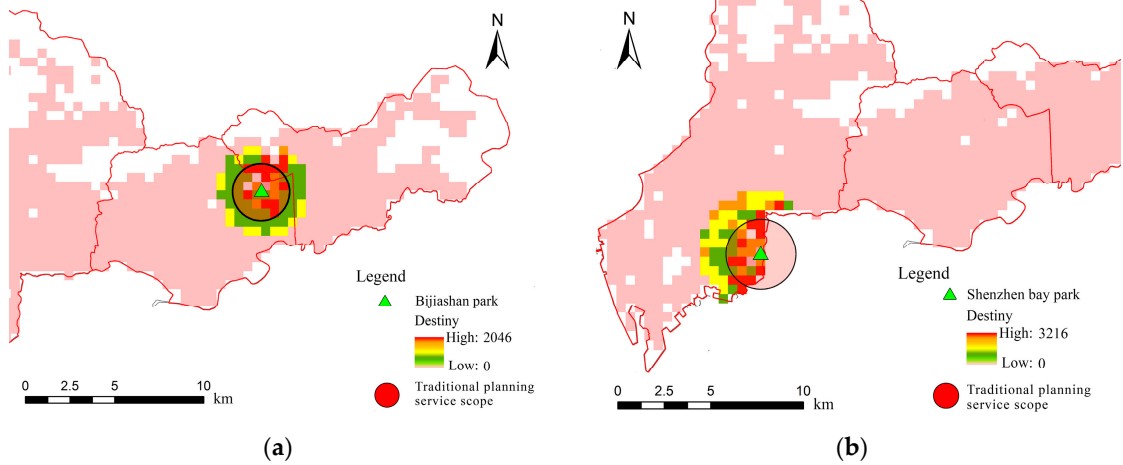

(**a**) (**b**)

**Figure 11.** A comparison of the traditional service scope and the service scope for big data visualization. (**a**) Bijiashan park and (**b**) Shenzhen bay park.

In China, traditional planning overemphasizes quantitative standards and simply increases the supply of urban parks without adequately considering the access mode of visitors. In the end, the best social effects are not obtained [39]. The study took into account the access mode of visitors. It was based on a large amount of personal tracking data which described the mobile mode of the visitors and the relationship between visitors and UPGSs. The new classification method relies on dynamic data generated by human activities, which can explain the concept of social perception. At the same time, the behavioral characteristics of human spatial activities were extracted from big data, which revealed the spatial and temporal distribution as well as the connection and process of socio-economic phenomena [20]. The new classification method investigated the similarity of UPGSs from the spatial interaction perspective. By considering the interaction of visitors and UPGSs, planners and managers can understand the characteristics of UPGSs more deeply.

Many scholars have studied the classification of UPGSs from different angles and methods. Some scholars [40] classified the urban park system of Phoenix in the United States based on data on various materials, land cover, and architectural features. Then, the composition and distribution

of the park network were revealed by comparing the park type and community social dimension. Some scholars [41] investigated parks in Denver and Colorado, the United States, based on accessibility, area, and park quality. They studied park types for children and adolescents, and they revealed environmental inequities based on low income and ethnicity. Some scholars [42] developed a park typology to discuss the relationship among park types, physical activity, and obesity. In the end, parks were divided into nine categories. Some scholars [43] inserted spatial logic to reclassify the existing park spaces in Phoenix, the United States, to affect the code reform and public investment decisions. This study optimized the allocation of UPGS resources. The studies above made great contributions to the classification of the UPGSs. However, Shenzhen is an immigrant city with a large floating population in China. That is a quite challenging factor for accurate data collection, let alone the acquisition of data on structure, income, and ethnicity that imply huge labor, financial, and time costs—especially for a city with a population as complex as that of Shenzhen. Therefore, the methods mentioned above were not suitable for this study. The taxi data contained data on the floating population, and it was dynamic and reflected real-time trends. Compared to traditional method, the cost of obtaining taxi data was small. However, the content attribute of taxi data was single, which was its limitation. The purpose of the classification of UPGSs is to enable urban residents have relatively fair access to green space resources and services. Most of the research focuses on this topic. This study was based on accessibility in order to discuss the spatial equity of UPGSs. It is a new preliminary study method. It only considers physical factors and ignores social factors, such as income, race, user preferences, history, and social background. This is the limitation of the study. Different regions, different national conditions, and different geographical environments will result in different classifications.

The quality of the park is closely related to the type of park. There are many indicators of park quality, such as scope, availability, age, culture, facilities, amenities, incivilities, aesthetics, facilities, and entertainment features [44–46]. These indicators can reveal the differences in parks for different income groups and different ethnic groups [44]. They can also assess investment in park facilities [45]. At the same time, they can also test the correlation with health behavior and outcomes [46]. Relatively large parks have a variety of facilities and usually receive more investment. Therefore, they have higher quality. However, most of studies above applied to the Western environment, especially the United States. The reason why these methods were not applicable was the lack of open data and cultural differences in developing countries. Although this study did not directly study park quality, it indirectly described the role of the park by the service scope. The larger the service scope value of the UPGS, the more attractive it was. The diverse services and high level facilities of large UPGSs make them attractive. Therefore, the more investment a park got, the higher its quality. However, due to the single attribute of taxi data, this study only qualitatively described the quality of the park. In future research, more park quality factors will be included to achieve a quantitative description of it.

The study did not consider the impact of population density on the spatial distribution of visitors. In addition, taxis data only covers a part of visitors, and the proportion of other travel modes, such as walking [47], bicycles [47], buses, subways, and private cars, is also high. In the future, we will use more data to study the relationship between visitors and green spaces. At the same time, we must consider more factors that affect the attractiveness of the park, such as the desirability of the service facilities, accessibility, mobility, and security of the UPGSs [48,49]. In the future, we hope to use a more efficient method to investigate the spatial layout and the allocation of management resources for UPGSs and provide guidance for urban planners and managers.

## 6. Conclusions

We used Shenzhen as a case study to investigate the spatial layout and management resource allocation for UPGSs based on taxi trajectory data from the spatial interaction perspective. The data were sourced from more than 16,000 taxis in Shenzhen for three months. We then calculated the percentage of the PUPs and DOPs in the buffer of the twenty two municipal parks of the three districts

and rasterized and visualized the results of the calculation. The service scopes of the UPGSs was quantified, thus describing the spatial distribution of the visitors and the intensity of people's activities. Subsequently, the spatial interaction was quantified by the distance attenuation, and a quadratic function was fitted to reveal the potential distance attenuation effect. Using the attenuation rate |a| as a comparative factor, the twenty two municipal parks were reclassified using the unsupervised classification method (k-means clustering). The results showed that the service scopes of the same type of UPGSs in the traditional classification were not the same as the one obtained by the study, but the attenuation rates of the same type were similar. The spatial distribution of visitors was related to the type and distance of the UPGSs. The distribution of visitors to theme parks (such as children's parks and children's playgrounds) was less affected by distance attenuation. The single function comprehensive parks were urban tourist attractions and they were sensitive to distance. In short, distance is an important factor for visitors when choosing UPGSs. However, distance is not the most important factor when people have special needs (such as family travel); instead, it becomes insensitive. The study will help planners improve the spatial distribution of UPGSs. At the same time, UPGSs were reclassified based on spatial interaction information to optimize the allocation of UPGS management resources. This method is complementary to traditional green space planning methods in China. Studying UPGS based on taxi trajectory data is a new research method and perspective. The study results are expected to provide better support for green space construction and management in Shenzhen.

**Author Contributions:** X.Z. is the project administration; he reviewed and edited the manuscript. Q.Z. conceived and designed the study; he also wrote the paper. M.J. and Q.Z. performed the experiments. All authors read and approved the manuscript.

**Funding:** This research study was funded by the National Natural Science Foundation of China (grant numbers 51438005,51878206).

**Conflicts of Interest:** The authors declare no conflict of interest.

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
