# Peer review of "Research on Urban Public Green Space Planning Based on Taxi Data: A Case Study on Three Districts of Shenzhen, China"

_sustainability, doi:10.3390/su11041132_

Round 1
Reviewer 1 Report
General comments
I have reviewed the manuscript and recommended for major revision. I am pleased with the effort that the authors made for improving their work. However, there is still room for further improvement. The main issue is the authors did address most of my previous comments, yet they had not been incorporated within the text. The language of the manuscript needs improvement by changing lengthy and complicated sentences. Additionally, the figures not in publication quality. Therefore. I, again, recommend the paper for major revision. For specifics please see the comments below.
* for my previous comments
Major comments
Urban green space can also mitigate urban heat island, which has negative impacts on the urban population. Also, highlight you this in your 1st para. of intro by citing following or similar papers.
Weng, Qihao, Dengsheng Lu, and Jacquelyn Schubring. "Estimation of land surface temperature–vegetation abundance relationship for urban heat island studies." Remote sensing of Environment 89.4 (2004): 467-483.
Maimatiyiming, M., Ghulam, A., Tiyip, T., Pla, F., Latorre-Carmona, P., Sawut, M., Halik, Ü., Caetano, M. (2014). Effects of spatial pattern of green space on land surface temperature: implications for sustainable urban planning and climate change adaptation. ISPRS Journal of Photogrammetry and Remote Sensing, 89: 59-66.
*Why 500 by 500 cells? You did address well in the response letter but it has to be reflected in the manuscript as well. Same for below comments.
*As we all know, the park shapes not a perfect square or a rectangle, in this case, how did you handle the distance difference from the center of the park to its surroundings? This is very important when looked at the attenuation index based on distance.
* justify why Jul.-Sep. were selected.
I think figure 10 is the same figure you have for figure 5a and b. plus, does not make a valid comparison of traditional and new service scope. I believe both are the service scope based on the big data.
Minor comments
Line 13, what is ‘P’ in ‘SCUPGS’?
Lines 50-53, long sentence, break down.
Line 89, specify ‘it’.
Lines 90-92, long sentence, break down.
On figure 7 and 9, letters too small too read.
In figure 4, municipal park represented with a circle and a dot in it, not just circle.
Author Response
Response to Reviewer 1 Comments
Point 1:Urban green space can also mitigate urban heat island, which has negative impacts on the urban population. Also, highlight you this in your 1st para. of intro by citing following or similar papers.
Response 1: Thanks for the comment. I modified the content in the first paragraph as follows: “Urban green space can also mitigate urban heat islands, which have a negative impacts on the urban population[3-4].” and added a reference.
Point 2:*Why 500 by 500 cells? You did address well in the response letter but it has to be reflected in the manuscript as well. Same for below comments.
Response 2: Thanks for the comment. The modified sentence is as follows: “Previously, 1000*1000 m and 200*200 m cells were tested separately. In the former case, because of the large scale, there were two or more green spaces in one cell. In the latter case, a green space was divided into a plurality of cells due to the small size.”
Point 3:*As we all know, the park shapes not a perfect square or a rectangle, in this case, how did you handle the distance difference from the center of the park to its surroundings? This is very important when looked at the attenuation index based on distance.
Response 3: Thanks for the comment. The modified sentence is as follows:“In addition, because the actual boundaries of the UPGSs were irregular shapes, data were difficult to obtain. If the actual boundary of the UPGS is considered, other factors such as its internal road network should also be considered. The complexity and scale of the study thus increased. The shape core of the UPGSs was chosen because many scholars [27-29] used it to study spatial accessibility and fairness. This is a mature research method, and the error of the result calculated by this method is within an acceptable range.”
Point 4:* justify why Jul.-Sep. were selected.
Response 4: Thanks for the comment. The modified sentence is as follows:“According to the “Analysis of Monthly Congestion Trends of Major Cities in 2017” [26], the data from July to September was chosen because the peak period of congestion in major cities in China begins to rise from July and peaks in September. During this period the city's transportation and tourism activities are the most representative. In addition, because the students' summer vacations is between July and September, their traffic frequency is higher than usual. ”
Point 5:I think figure 10 is the same figure you have for figure 5a and b. plus, does not make a valid comparison of traditional and new service scope. I believe both are the service scope based on the big data.
Response 5: Thanks for the comment. I modified the content of Figure 10 to make the contrast between them clearer.
|
Point 6:Line 13, what is ‘P’ in ‘SCUPGS’?
Response 6: Thanks for the comment. I am very sorry, this is a mistake. I confused UPGS and SCUGS when I was writing. I have removed the wrong part.
Point 7:Lines 50-53, long sentence, break down.
Response 7: Thanks for the comment. I broken down the sentence. Since the new normative standard was just implemented in China in 2018, the content was revised. Although the new specification adds some content, it has no impact on the research. The modified sentence is as follows: “However, in China the planning, classification and management of green space are based on the Standard for Classification of Urban Green Space (SCUGS) by the Ministry of Construction in 2018. SCUGS is based on the service radius and rigid indices such as the green space area per capita and the green space ratio, etc., to guide urban planning.”
Point 8:Line 89, specify ‘it’.
Response 8: Thanks for the comment. The modified sentence is as follows:“Therefore, this work studied the relationship between people and cities based on big data and a "human-oriented" perspective.”
Point 9:Lines 90-92, long sentence, break down.
Response 9: Thanks for the comment. I broken down the sentence.“The study took the three districts of Shenzhen as an example, and used taxi trajectory data to investigate the spatial layout of UPGSs and manage resource allocation from the spatial interaction perspective. The method was used to compensate for the limitations of traditional planning methods in China. ”
Point 10:On figure 7 and 9, letters too small too read.
Response 10: Thank you very much for your comments, I modified the size and format of the image.
Point 11:In figure 4, municipal park represented with a circle and a dot in it, not just circle.
Response 11: Thank you very much for your comments, I have modified the error of the picture.

Reviewer 2 Report
This manuscript is significantly improved. Thank you for your thoughtful responses and revisions. There are still a few items that cause concerns and need to be addressed. See below – the numbers reference those in your response.
The main issue that needs to be addressed is the 200m buffer around green spaces - which seems too large to actually capture green space visits only (vs. visits to nearby shops or restaurants). I would like to see a much more thorough explanation for that, and possibly, the comparison of results between 200m and other buffers (say, 100 m).
2. It still not clear what the literature gap is. Has anybody ever analyzed the shortcomings of SCUPGS? (Innovation for the evaluation of a planning approach). Or has anybody ever used taxi data to describe access to parks, or are you introducing new techniques to use taxi data? (Methodological innovation). Please clarify the gaps and your contributions in the abstract, introduction, and conclusion.
7. Taxi data describes residents’ travel patterns. You do not know whether the parks the visit actually meet their needs. You are not asking them whether they are satisfied with the parks they visit. For example, people might be visiting a park because it is convenient to them, not because it meets all their needs. When I choose the parks I visit, I find some middle ground between what is accessible and what my needs are. Of course, I would love to visit a place like Central Park in New York City every day, but I do not live there. Also, I might feel like I need to visit a park every day, but I do not have the time to do so. So my limited visits to a park do not meet my need because I would like to go more often but work and family duties get in the way. Therefore, I strongly suggest that you rephrase the concept of “access need” with “visitation patterns” or something similar. Your data is about park use and not about whether needs are met.
12. I suggest rephrasing the sentence as, “Therefore, UPGSs are receiving increasing attention in numerous research fields, such as ecology and geography [11,12,13].”
14. Reconsider writing the last sentence with the active voice. Who is not considering residents needs? Planners? Policymakers?: “In some circumstances, planners and policymaker study UPGSs using a top-down approach and, therefore, do not fully consider the actual needs of residents.”
18. Thank you for the thorough explanation regarding the share of taxi trips to parks. Below are a few suggestions, which mostly deal with the fact that, in the revised paper, you do not mention any of these data. Therefore, a reader is still left to wonder why using taxi data matters, as you do not report any of the percentages of trips to parks done by taxi found in study [2] cited in your response (“Sustainability of Recreational Travel to Parks in Chinese Metropolitan Areas: Case Study in Shanghai”) or the results of the questionnaire you conducted.
Suggestion 1. In the introduction, make the case that it matters to study park visits with a taxi mode because, at least in some cities in China, taxi is not a small for of travel to parks – and cite study [2.]
Suggestion 2. In Section 2 (“Research Data Description and Processing”), you should report the findings of study [2] cited above more in detail. Also, you should report details about the methods and the results of your questionnaire, which, to some extent, support your choice to use taxi data to study park visits.
Suggestion 3. Based on your questionnaire, the share of people who use taxis to parks is still relatively small – actually, the smallest after private car. This is, in some way, a limitation of using taxi data. Based on Figure 2 above, it would make more sense to study subway trips to parks if data were available. You should note that the low share of taxi riders to parks – compared to other modes – is a limitation in itself, and you should report it in Section 2 and then again in Section 6 (Conclusion) when you report the limitations.
19. A 200m buffer still seems very large. Again, that might describe trips to other destinations unless the streets near parks are 200m wide, which would seem like too much even for a very large highway. I do not understand your explanation saying, “Since the size of the municipal park is large enough, the distance between its boundary and the buffer zone is in the range of 30-50 m.” Based on that, wouldn’t you want to use a buffer of, say, 70-80m maximum? A zoomed-in map of park boundaries, streets, and data points (drop off and pick up) would be helpful to understand why you chose 200m – which, I think, is still too large. Please include this map in the revised paper.
24. Thank you for the clarifications and edits. Because this new classification is an outcome of your paper, I suggest that you cover it more extensively in the discussion (Section 5). Specifically, you could compare your classification to others presented in studies in other geographical contexts. This is particularly important because the journal has an international audience. For other park classifications, see, for example (Bird et al., 2016; Ibes, 2015; Jones, Brainard, Bateman, & Lovett, 2009; Rigolon, 2017; Talen, 2010). Takin the discussion one step further, you could talk about park quality, which is an important element closely linked to park types. Indeed, larger parks tend to have the largest variety of amenities and, often, receive more investment – thus, they have higher quality. For relevant studies defining park quality, see (Broomhall, Giles-Corti, & Lange, 2004; Edwards et al., 2013; Hughey et al., 2016; Kaczynski et al., 2016; Rigolon & Németh, 2018; Vaughan et al., 2013).
References
Bird, M., Datta, G. D., van Hulst, A., Cloutier, M. S., Henderson, M., & Barnett, T. A. (2016). A park typology in the QUALITY cohort: Implications for physical activity and truncal fat among youth at risk of obesity. Preventive Medicine, 90, 133–138. https://doi.org/10.1016/j.ypmed.2016.06.042
Broomhall, M. H., Giles-Corti, B., & Lange, A. (2004). Quality of Public Open Space Tool (POST). Retrieved January 2, 2016, from http://www.see.uwa.edu.au/research/cbeh/projects/post
Edwards, N., Hooper, P., Trapp, G. S. A., Bull, F., Boruff, B., & Giles-Corti, B. (2013). Development of a Public Open Space Desktop Auditing Tool (POSDAT): A remote sensing approach. Applied Geography, 38, 22–30. https://doi.org/10.1016/j.apgeog.2012.11.010
Hughey, S. M., Walsemann, K. M., Child, S., Powers, A., Reed, J. A., & Kaczynski, A. T. (2016). Using an environmental justice approach to examine the relationships between park availability and quality indicators, neighborhood disadvantage, and racial/ethnic composition. Landscape and Urban Planning, 148, 159–169. https://doi.org/10.1016/j.landurbplan.2015.12.016
Ibes, D. C. (2015). A multi-dimensional classification and equity analysis of an urban park system: A novel methodology and case study application. Landscape and Urban Planning, 137, 122–137. https://doi.org/10.1016/j.landurbplan.2014.12.014
Jones, A. P., Brainard, J., Bateman, I. J., & Lovett, A. A. (2009). Equity of access to public parks in Birmingham, England. Environmental Research Journal, 3(2/3), 237–256. Retrieved from http://cat.chem.chiba-u.jp/PDF/2009EnvironResJ.pdf#page=99
Kaczynski, A. T., Schipperijn, J., Hipp, J. A., Besenyi, G. M., Wilhelm Stanis, S. A., Hughey, S. M., & Wilcox, S. (2016). ParkIndex: Development of a standardized metric of park access for research and planning. Preventive Medicine, 87, 110–114. https://doi.org/10.1016/j.ypmed.2016.02.012
Rigolon, A. (2017). Parks and young people: An environmental justice study of park proximity, acreage, and quality in Denver, Colorado. Landscape and Urban Planning, 165, 73–83. https://doi.org/10.1016/j.landurbplan.2017.05.007
Rigolon, A., & Németh, J. (2018). A QUality INdex of Parks for Youth (QUINPY): Evaluating urban parks through geographic information systems. Environment and Planning B: Urban Analytics and City Science, 45(2). https://doi.org/10.1177/0265813516672212
Talen, E. (2010). The spatial logic of parks. Journal of Urban Design, 15(4), 473–491. https://doi.org/10.1080/13574809.2010.502335
Vaughan, K. B., Kaczynski, A. T., Wilhelm Stanis, S. A., Besenyi, G. M., Bergstrom, R., & Heinrich, K. M. (2013). Exploring the distribution of park availability, features, and quality across Kansas City, Missouri by income and race/ethnicity: An environmental justice investigation. Annals of Behavioral Medicine, 45(Suppl. 1), S28–S38. https://doi.org/10.1007/s12160-012-9425-y
26. I would like to reiterate that the park use you measure through taxi data does not say much about park need. Please remove any reference to park need.
Author Response
Response to Reviewer 2 Comments
Point 1:It still not clear what the literature gap is. Has anybody ever analyzed the shortcomings of SCUPGS? (Innovation for the evaluation of a planning approach). Or has anybody ever used taxi data to describe access to parks, or are you introducing new techniques to use taxi data? (Methodological innovation). Please clarify the gaps and your contributions in the abstract, introduction, and conclusion.
Response 1:Thank you very much for your suggestion. My revised content is as follows:Although many scholars use taxi trajectory data to study public service facilities, few scholars use taxi trajectory data to describe UPGSs.
Point 2:Taxi data describes residents’ travel patterns. You do not know whether the parks the visit actually meet their needs. You are not asking them whether they are satisfied with the parks they visit. For example, people might be visiting a park because it is convenient to them, not because it meets all their needs. When I choose the parks I visit, I find some middle ground between what is accessible and what my needs are. Of course, I would love to visit a place like Central Park in New York City every day, but I do not live there. Also, I might feel like I need to visit a park every day, but I do not have the time to do so. So my limited visits to a park do not meet my need because I would like to go more often but work and family duties get in the way. Therefore, I strongly suggest that you rephrase the concept of “access need” with “visitation patterns” or something similar. Your data is about park use and not about whether needs are met.
Response 2:Thank you very much for your suggestion. I modified the “access need” to “access mode”.
Point 3:I suggest rephrasing the sentence as, “Therefore, UPGSs are receiving increasing attention in numerous research fields, such as ecology and geography [11,12,13].”
Response 3:Thank you very much for your suggestion. I modified the sentence according to your suggestion.
Point 4:Reconsider writing the last sentence with the active voice. Who is not considering residents needs? Planners? Policymakers?: “In some circumstances, planners and policymaker study UPGSs using a top-down approach and, therefore, do not fully consider the actual needs of residents.”
Response 4: Thanks for the suggestion, I modified the content based on your suggestions.
Point 5:Thank you for the thorough explanation regarding the share of taxi trips to parks. Below are a few suggestions, which mostly deal with the fact that, in the revised paper, you do not mention any of these data. Therefore, a reader is still left to wonder why using taxi data matters, as you do not report any of the percentages of trips to parks done by taxi found in study [2] cited in your response (“Sustainability of Recreational Travel to Parks in Chinese Metropolitan Areas: Case Study in Shanghai”) or the results of the questionnaire you conducted.
Suggestion 1. In the introduction, make the case that it matters to study park visits with a taxi mode because, at least in some cities in China, taxi is not a small for of travel to parks – and cite study [2.]
Suggestion 2. In Section 2 (“Research Data Description and Processing”), you should report the findings of study [2] cited above more in detail. Also, you should report details about the methods and the results of your questionnaire, which, to some extent, support your choice to use taxi data to study park visits.
Suggestion 3. Based on your questionnaire, the share of people who use taxis to parks is still relatively small – actually, the smallest after private car. This is, in some way, a limitation of using taxi data. Based on Figure 2 above, it would make more sense to study subway trips to parks if data were available. You should note that the low share of taxi riders to parks – compared to other modes – is a limitation in itself, and you should report it in Section 2 and then again in Section 6 (Conclusion) when you report the limitations.
Response 5: Your suggestion is very helpful for the modification of the article. According to your suggestion, the changes are as follows: In the Section 2, a paragraph is added.“Taxi mode is one of the important modes of visiting UPGSs in some cities in China. Tianqing et al. [25] selected eleven well-known UPGSs in Shanghai. The study was completed in the sunny days of March–June 2009. A total of 15 surveys were conducted and 588 valid questionnaires were obtained. The study found that four UPGSs had a higher proportion of non-sustainable travel modes (taxis) than the other parks, with the proportion of non-sustainable travel modes for the four parks being 24%, 37%, 40% and 45%. This shows that the proportion of taxi mode access to UPGSs is not low. In addition, in order to demonstrate the proportion of taxi mode travel, we selected six municipal parks in the three districts of Shenzhen for a questionnaire survey. When studying UPGSs with similar distances, one should pay attention to the difference in size and completion time to increase the representativeness of the sample. The investigation of the municipal parks was concentrated in the sunny days of October 2018. We collected data on the weekend when the visit rate was high. The questionnaire included the travel mode and source of the visit. A total of five surveys were conducted and 1021 valid questionnaires were obtained. The survey found that 11.36% of visitors chose to visit the UPGSs in taxi mode. The growth of the taxi mode stems from the development of the network car. However, the most popular modes of travel were still subway (32.81%) and bus (19.59%), followed by walking (16.16%) and bicycles (11.66%). Private cars were only (8.42%). Although the taxi mode has grown faster, the proportion of taxis is still small compared to other travel modes. Due to the relatively small proportion of taxi users, the study has some limitations. However, the population has a large base in Shenzhen, and the taxi model is still very valuable for study.”
This content is re-emphasized in the conclusion section.“ In addition, taxis data only covers a part of UPGSs visitors, and the proportion of other modes of travel is also high, such as walking [47], bicycles [47], buses, subways and private cars. In the future, we will use more data (such as shared bicycle data) to study the relationship between visitors and green spaces. ”
Point 6:A 200m buffer still seems very large. Again, that might describe trips to other destinations unless the streets near parks are 200m wide, which would seem like too much even for a very large highway. I do not understand your explanation saying, “Since the size of the municipal park is large enough, the distance between its boundary and the buffer zone is in the range of 30-50 m.” Based on that, wouldn’t you want to use a buffer of, say, 70-80m maximum? A zoomed-in map of park boundaries, streets, and data points (drop off and pick up) would be helpful to understand why you chose 200m – which, I think, is still too large. Please include this map in the revised paper.
Response 6: Thank you very much for your comments. Your comments have been very helpful in modifying this issue. The content of the article is revised as follows: “ As shown in Figure 3, the percentage tends to be smooth starting at 200m, which is a stable interval. A threshold of 200m was used, which is an acceptable walking distance for a person. A study found that the distance between the PUPs (or DOPs) of the taxi and the actual place of departure (or the place of arrival) is mostly around 200 m [23]. Since the environment around municipal parks is very different, it was actually impossible to completely distinguish between different situations. In order to include more possibilities, we chose a larger buffer. Following this, we drew a 200m buffer for each UPGSs, which ensured that there was no interaction between each buffer (Figure 4). Although the 200m buffer caused some noise, there were many ways to eliminate the noise. For example, because the taxi data was all-day data, we used the time dimension to select the data from the morning and evening peaks, as well as the data for meal times to exclude noise from residential areas, schools and restaurants. In addition, commercial, medical and other functions near UPGSs are interactive, and some proportion of visitors take the opportunity to go to the UPGSs for leisure. In summary, although the data had some noise, the sample size was large, and the proportion of noise in the sample size was very small. Therefore, the error was within an acceptable range. ”
First, as shown in Figure 4, the surrounding environment of the twenty-two parks studied varies greatly. Some are surrounded by mountains and lakes, some are surrounded by the sea, some are surrounded by viaducts and rivers, and some are in the city center. In real life, due to the diversion of people and vehicles and traffic control, the actual boarding and getting off points of taxis are not the entrances and exits of the park. People need to walk to a place where it is easier to get a taxi to get service. Therefore, 200m is an acceptable walking range for pick-up and drop-off points and actual departures and destinations[1-2]. But if the buffer is too small, the possibility of many actual visits will be ignored.In order to include more access possibilities, we choose a larger buffer.
Secondly, residential areas, schools, restaurants, etc. within the 200m buffer range have a fixed time interval. Since the taxi data is all-day data, we can filter the noise through the time dimension, which is very low in proportion to the sample size.
Third, the proportion of large shopping malls and hospitals around the twenty-two parks studied is not high. In addition, because these functions complement the functions of UPGSs, people interact with UPGSs when accessing such facilities, and the green space is also accessed (In the questionnaire, 68% of visitors will visit the park by the way). The buffer zone of about 100m radiates less than other complementary facilities, and the expression of characteristic is not as good as 200m.
Finally, in future research, we will study from the green areas of the mesoscale and microscale with typical characteristics. Considering more influencing factors, the noise of the data can be reduced by selecting an appropriate buffer to better study the characteristics of the green space. Your suggestion is very important and we will include your suggestion in the next study.
References:
[1] Kong, X., Liu, Y., Wang, Y., Tong, D., Zhang, J. Investigating public facility characteristics from a spatial interaction perspective: a case study of Beijing hospitals using taxi data. ISPRS International Journal of Geo-Information. 2017, 6(2), 38.
[2] Gong, Li , et al. "Inferring trip purposes and uncovering travel patterns from taxi trajectory data." Cartography and Geographic Information Science. 2015, 43.2:1-12.
Point 7:Thank you for the clarifications and edits. Because this new classification is an outcome of your paper, I suggest that you cover it more extensively in the discussion (Section 5). Specifically, you could compare your classification to others presented in studies in other geographical contexts. This is particularly important because the journal has an international audience. For other park classifications, see, for example (Bird et al., 2016; Ibes, 2015; Jones, Brainard, Bateman, & Lovett, 2009; Rigolon, 2017; Talen, 2010). Taking the discussion one step further, you could talk about park quality, which is an important element closely linked to park types. Indeed, larger parks tend to have the largest variety of amenities and, often, receive more investment – thus, they have higher quality. For relevant studies defining park quality, see (Broomhall, Giles-Corti, & Lange, 2004; Edwards et al., 2013; Hughey et al., 2016; Kaczynski et al., 2016; Rigolon & Németh, 2018; Vaughan et al., 2013).
Response 7: Thank you very much for your comments. According to your suggestion, we compare our classification to others presented in studies in other geographical contexts.And we discuss about park quality, the revised content is as follows:“ Many scholars have studied the classification of UPGSs from different angles and methods. Some scholars [40] classified the urban park system of Phoenix in the United States based on data on various materials, land cover and architectural features. Then, the composition and distribution of the park network were revealed by comparing the park type and community social dimension. Some scholars [41] surveyed parks in Denver and Colorado, USA, based on accessibility, area and park quality, and then studied park types for children and adolescents, revealing environmental inequities based on low income and ethnicity. Some scholars [42] developed a park typology to discuss the relationship between park type, physical activity and obesity, finally dividing parks into nine categories. Scholars [43] inserted spatial logic to reclassify the existing park space in Phoenix, USA, affecting code reform and public investment decisions. This study optimized the allocation of UPGS resources. The above studies made great contributions to the classification of UPGSs. However, Shenzhen is an immigrant city in China with a large floating population. Accurate data on the floating population is difficult to obtain. Obtaining accurate data on structure, income, and ethnicity requires huge human, financial, and time costs. It is a huge difficulty for a city with a population as complex as that of Shenzhen. Therefore, the methods of the above studies were not suitable for this study. The taxi data contained data on the floating population, which was its advantage. Currently, most of the research was based on static data. The taxi data was dynamic data and reflected real-time trends. Compared to traditional data, the advantage of taxi data is that the cost of acquisition is small. However, the content attribute of taxi data is single, which is its limitation. The purpose of the classification of UPGSs is to enable urban residents have relatively fair access to green space resources and services. Most of the research is centred around this topic. Compared with the above studies, this study was based on accessibility in order to discuss the spatial fairness of UPGSs. This study is merely a new preliminary method, it considers physical factors, not social factors like income, race, user preferences, history and social background. This is the limitation of the study. Different regions, different national conditions, and different geographical environments will result in different classifications.
The quality of the park is closely related to the type of park. There are many indicators of park quality, such as scope, availability, age, culture, facilities, amenities, incivilities, aesthetics, facilities and entertainment features [44-46]. These indicators can reveal the differences in parks for different income groups and different ethnic groups [44]. It can also assess investment in park facilities [45]. At the same time, it can also test the correlation with health behaviour and outcomes [46]. Relatively large parks have a variety of facilities and usually receive more investment. Therefore, they have higher quality. However, most of the above studies apply to the Western environment, especially the United States. The reason why these methods are not applicable is the lack of open data and cultural differences in developing countries. Although this study did not directly study park quality, by being based on taxi data it indirectly describes the use of the park by the service scope. The larger the service scope value of the UPGS, the more attractive it is. The diverse service offerings and high level of facilities of large UPGSs make them attractive. Therefore, they get more investment and the quality of the park is higher. This is a benign cycle. However, due to the single attribute of taxi data, income, ethnicity, culture and other factors are not considered. Therefore, this research could only qualitatively describe the quality of the park. In future research, more park quality factors will be included to achieve a quantitative description of the park quality.”
Point 8: I would like to reiterate that the park use you measure through taxi data does not say much about park need. Please remove any reference to park need.
Response 8: Thank you very much for your comments. According to your suggestion, we removed the reference to park need.

Round 2
Reviewer 1 Report
I greatly appreciate the authors' effort in improving the quality of their manuscript. I believe the manuscript is ready to be published after a few minor changes.
For consistency, update the Chinese to English in Fig4.
Make the traditional planning service scope in Fig 10 transparent to the degree that background can be seen. 40-20% transparency may work.
Always check for spelling and grammatical errors throughout.
Author Response
Thank you very much for your guidance. Under your careful guidance of the article, the article becomes rigorous and the subject becomes clear. Thank you for your reference, which has enriched the content of the article. These suggestions and comments have given me a lot of gains. I have checked the entire article several times and the previous question has been revised. Thank you again for your help.

Reviewer 2 Report
I appreciate the extensive edits that you made and the thoughtful responses to my comments. I do not have any additional substantial suggestions and I am happy to suggest accepting the paper after very minor revisions. My only comment is that you will need to proofread the manuscript and clean up some grammar mistakes and typos. For example, you wrote “taxi model” twice (once as the heading for section 2.2). What I think you actually meant to write is “taxi travel mode.” Indeed taxi is a transportation mode, or mode of transportation. Also, on page 12, line 362, you wrote, “The distribution of the distribution of the UPGS…”. Clearly, this is a repetition that needs to be eliminated. There are many others of these small issues throughout the manuscript, and those need your attention to meet the standard of quality of Sustainability.
Author Response
Thank you very much for your guidance. Under your careful guidance of the article, the article becomes rigorous and the subject becomes clear. Thank you for your reference, which has enriched the content of the article. These suggestions and comments have given me a lot of gains. I have checked the entire article several times and the previous question has been revised. Thank you again for your help.

This manuscript is a resubmission of an earlier submission. The following is a list of the peer review reports and author responses from that submission.
Round 1
Reviewer 1 Report
This paper looks at travel to and from urban parks with taxi data. The authors use big data from taxi trips in a Chinese city. I have several concerns that need to be addressed to improve the paper and bring it to the standards of Sustainability. First, the research purpose is unclear and the research questions are missing. Second, claiming that planners “ignore the actual needs of users” is not true (at least in many Global North cities), which undermines most of your argument to write this paper. Third, I am skeptical that studying people traveling to parks via taxi is a worthwhile endeavor, as you do not provide any evidence on the percentage of people traveling to parks via taxi in the studied city (or in general). Based on studies in the U.S., Australia, and China, this percentage is very low. Four, I am not convinced about the choice of a 270 m (or 200 m, unclear) buffer to operationalize the trips ending at parks; such a large distance might pick up significant “noise” of people traveling to nearby restaurants, stores, etc. Finally, I do not find any clarification about how taxi data can help planners identify the needs of park users, which is a central piece of your argument. Below are more detailed comments organized by section.
Title
- There is no need to mention the name of the districts in the title. This journal has an international readership, and most people are unfamiliar with the name of those districts. You can just say “A case study on three districts of Shenzhen, China.”
Abstract
- The abstract should include the research gap and justification for this study. What literature gap are you filling? How are you contributing to the scientific literature, and possibly to practice? In other words, what is “new” about your study?
- The purpose of the study is unclear and specific research questions are missing. In the abstract, you say, “The purpose of this study is to examine the differences in the spatial characteristics of different types of UPGS, from the perspective of spatial interaction.” What are the “spatial characteristics?” You study travel to UPGS, but “spatial characteristics” make me think about the design of parks. In the introduction, you do not clarify the purpose of the study nor report specific research questions.
- What is “The traditional UPGS planning method?” Is it a top-down approach to planning parks and green spaces in which the resident’s voices are not heard? In the U.S. and most of Europe, most planning processes are participatory, i.e., they include the needs of residents and visitors. When you refer to “The traditional UPGS planning method,” are you referring to a specific planning issue in Chinese cities?
- Again, there is no need to mention the three neighborhoods by name in the abstract.
- “based on spatial interaction extracted from taxi data.” Do you mean “spatial interaction data?” Also, what is taxi data? Taxi cabs? Or ride-sharing companies like Uber? I know the abstract can only include so many words, but this needs to be clarified because it’s, from my reading, the novel part of your approach.
- In the abstract, I also struggle to connect the need to include visitors’ needs in planning for UPGS with the use of taxi data. What does taxi data tell us about visitors’ needs related to UPGS? Also, are taxi data accessible to the average UPGS planner in China (and other contexts)?
- It is not correct to start a sentence with a number that is not written out. Thus, rather than “22 municipal parks,” please write “Twenty-two municipal parks.”
- What is the “distance decay effect?” Overall, you need to write your abstract – and most of the paper content, with some exceptions in the methods section – in more comprehensible language that every researcher can understand. In other words, you need to write for a broader audience.
Introduction
- The text needs substantial copy-editing. For example, the first sentence could be rephrased as: “Urban public green space (UPGS), which provide natural environments and public spaces in cities, have numerous benefits that are widely recognized.” In the second sentence, use “making cities more attractive” – not “the city,” which would imply you have a specific city in mind. I will stop here with comments on copy-editing, but there are many more issues.
- The first few sentences should include a brief explanation of what you mean by UPGS: Parks, tree canopy, and/or other privately-owned green spaces?
- You say: “Thus, UPGSs are receiving increasing attention from several research fields. The core view of these studies is to assess the link between the supply positions and demand positions from a spatial perspective.” Which research fields and studies are you referring to? Please provide citations.
- The following statement is highly problematic, and it makes me question the overall argument of your manuscript: “The traditional UPGS planning is based on the service radius and combines the subjective experience of urban planners and managers to allocate UPGSs, from the supply side. Urban planners and managers only emphasise the indicators and supply of quantity of the UPGS but ignore the actual needs of users [9].” First of all, claiming that all urban planners “ignore the actual needs of users” seems like a very harsh statement. You mean that planners do not care? Do they ever engage with constituents? In many Global North countries, they do. It is a key part of their job. Also, the citation you use to back this up is from 1999, which is almost 20 years ago. I doubt that anybody, today, would claim based on empirical evidence that urban planners deliberately “ignore the actual needs of users.” Planners still make some decisions with a top-down approach, but public outreach and participation are now required in many countries around the world.
- There are two alternative solutions to the issue raised in the previous comment. First, you can completely change the main argument in the abstract and introduction and find a different justification for your analysis. Alternatively, you can provide much stronger evidence that, in 2018, and perhaps only in China or some of its cities, planners “ignore the actual needs of users.” Also, I would make those statements less conclusive and say something like planners “in some circumstances use top-down approaches that do not fully consider the needs of users.”
- Also, I cannot find citation [9] in any literature database I can access. Is it published in Chinese only? I am asking because you based most of your argument on that citation, and, as a reader, I would like to double-check how the authors of that paper made the claim that planners “ignore the actual needs of users.”
- The following statement is also highly problematic and not true: “Few scholars study UPGSs from the perspective of visitors.” In fact, there have been many studies in the US, Europe, and China on the experiences and needs of people in urban green spaces such as parks. See journals like “Environment and Behavior,” “Landscape and Urban Planning,” and many others. Please remove that sentence and any other claim on the lack of studies on visitors’ perspectives of UPGSs.
- Not sure that the information included in lines 67-82 is necessary for the introduction. You can explain the methods in the following section. You should report the purpose and research questions of your study, but there is no need to go in-depth with the methods.
- This is another important point. Do you have any data on what percentage of park visitors use taxicabs to go to a park from their home? Based on the available research in the U.S., Australia, and China, such percentage is very small (Loukaitou-Sideris, & Stieglitz, 2002; Loukaitou-Sideris & Sideris, 2009; Veitch et al., 2014: Wang et al., 2014). In other words, if very few people use taxis to travel to parks (like these studies show), I question whether it is worth it to use taxi data to study the spatial relationship between park location and people’s homes. In other words, how representative are the people taking taxis to parks of the entire population? Do they live farther from parks than people who walk to parks? Overall, I am really not convinced that we should study access to parks with taxi data, and you need to make a much stronger argument (with supporting data about the share of people using taxis to parks) to support the study you conducted.
Research Data
- You use 270 m as a buffer around each park to identify taxi trips that have parks as destinations. How do you know that people who get dropped off in that buffer area are actually park visitors? That is a pretty big assumption to make because, especially in dense cities like the one you are studying, people might be going to a lot of other destinations located next to that park (e.g., a restaurant, a supermarket). I am really not convinced you can claim that all trips ending within a 270 m buffer can be considered as trips to parks. To solve this issue, I suggest limiting the buffer to a much smaller radius (e.g., 30-50 m) to make sure that the trips are counting actually end at a park’s perimeter.
Methods
- Section 3.1 repeats some of the information reported in the Introduction. I suggest eliminating those repetitions.
- It is unclear whether the K classes created in the “K-means clustering method” are related to the classification of UPGS provided by the Chinese government. Please clarify how the two relate, if at all. If they do not relate, more information is needed to justify the K-means clustering method. What research question are you answering through that method?
Results and Analysis
- Which criteria did you use to select the six parks? Did you randomize the choice? Or is it a purposive sampling approach based on the park classification? If it was the latter (purposive), how did you choose the two parks in each group? This information should be moved to the methods section.
- I am confused about whether you actually use six or 22 parks. The findings you report in section 4 mostly focus on the six parks – except the classification. Please clarify this duality in the methods and results section.
- Perhaps because the research questions are not included, I struggle to follow most of section 4. For example, why are you developing a “New classification of UPGS based on spatial interaction?” This new classification was not described as a key element of your paper in the introduction. I do not understand the utility of this classification, nor how it helps planners take into account visitors’ needs (which is one of your key justifications for using taxi data).
Discussion and conclusion
- A “bottom-up” approach would require residents to have a voice in planning parks and green spaces. Using taxi data cannot be described as “bottom-up.” Please remove this expression.
- In this section, I do not find any clarification about how taxi data can help planners identify the needs of park users. Such data do not provide information on people’s preferences for any design elements of parks nor do they tell us what people like to do when they visit parks.
References
Loukaitou-Sideris, A., & Stieglitz, O. (2002). Children in Los Angeles parks: A study of equity, quality and children’s satisfaction with neighbourhood parks. Town Planning Review, 73(4), 467–488. https://doi.org/10.3828/tpr.73.4.5
Loukaitou-Sideris, A., & Sideris, A. (2009). What brings children to the park? Analysis and measurement of the variables affecting children’s use of parks. Journal of the American Planning Association, 76(1), 89–107. https://doi.org/10.1080/01944360903418338
Veitch, J., Carver, A., Hume, C., Crawford, D., Timperio, A. F., Ball, K., & Salmon, J. (2014). Are independent mobility and territorial range associated with park visitation among youth? The International Journal of Behavioral Nutrition and Physical Activity, 11(1), 73. https://doi.org/10.1186/1479-5868-11-73
Wang, D., Brown, G., Zhong, G., Liu, Y., & Mateo-Babiano, I. (2015). Factors influencing perceived access to urban parks: A comparative study of Brisbane (Australia) and Zhongshan (China). Habitat International, 50, 335–346. https://doi.org/10.1016/j.habitatint.2015.08.032
Reviewer 2 Report
General comments
Taking advantage of taxi data, the manuscript by Zheng and colleagues studied the interactions between the visitors and urban public green space (UPGS). Overall, the topic is relevant and, analysis has some value for the field of urban planning and green space management. That said, the manuscript is structured poorly, the importance of the work is not well-highlighted, and the discussion is dull due to the insufficient literature review. Therefore, the manuscript is not ready and I recommend major revision. For further comments, please see below.
Major comments
In that abstract, you mentioned ‘the spatial characteristics of UPGS …’. In the manuscript, you did not look at the spatial aspect, but only park types. Also, lines ’24-26’, which are not totally true based on your conclusion.
Lines 48-49 must be expanded and discussed to highlight the unique contribution of your work.
Para 3 and 4 of the intro can be merged and do the same as above for lines 60-61.
Lines 62-63 can be moved to the first paragraph.
Big data related studies not well-reviewed.
Lines 68-82 not belong to the intro. Maybe for results/conclusion.
Make a separate paragraph for the objectives in the intro.
Two first paras of ‘POI data’ section is rather confusing This can be removed. What’s Gaode map? It sounds like you just located your POI on a digital using coordinates. You never used area and volume, did you?
Why 500 by 500m cells?
As we all know, the park shapes not a perfect square or a rectangle, in this case, how did you handle the distance difference from the center of the park to its surroundings? This is very important when looked at the attenuation index based on distance.
Specify exact source of the taxi data.
Line 121, justify why Jul.-Sep. were selected.
Line 124, specify what ‘preprocessing’ methods were used.
In section 3.1, specify what service scope values mean. Without this your description in the results rather confusing.
In section 3.2 it is not clear that how that 3 class types were determined.
Section 4.1 for does not make sense you only talked about population density while refereeing the service scope map (fig.4)
Lines 229-255 refer figure numbers in the text while the relevant description is presented.
There is no prior information about attenuation rate index until it is mentioned in the results, which makes your reader lose interest. Please explain it in the methods.
Separate the discussion from the conclusion and make the discussion more interesting by including similar studies, comparing them to yours and discussing your results thoroughly.
Minor comments
Fig 1. add map elements e.g., legend, north arrow, and scale etc.
Line 98-99, when you report what, you did use simple past tense, instead of present perfect tense. Correct throughout the text.
Add a reference for K-mean clustering.
Line 162, specify short names when they first appear.